# Rap1 prevents colitogenic Th17 cell expansion and facilitates Treg cell differentiation and distal TCR signaling

Sayaka Ishihara[1], Tsuyoshi Sato[1], Noriyuki Fujikado[2], Haruka Miyazaki[1], Takayuki Yoshimoto [3], Hiromitsu Yamamoto[4], Shinji Fukuda [4,5,6] & Koko Katagiri [1✉]

T-cell-specific Rap1 deletion causes spontaneous colitis in mice. In the present study, we revealed that Rap1 deficiency in T cells impaired the preceding induction of intestinal RORγt$^+$ Treg cells. In the large intestinal lamina propria (LILP) of T-cell-specific Rap1-knockout mice (Rap1KO mice), Th17 cells were found to increase in a microbiota-dependent manner, and the inhibition of IL-17A production prevented the development of colitis. In the LILP of Rap1KO mice, RORγt$^+$ Treg cells were scarcely induced by 4 weeks of age. The expression of CTLA-4 on Rap1-deficient Treg cells was reduced and the expression of CD80 and CD86 on dendritic cells was consequently elevated in Rap1KO mice. When cultured under each polarizing condition, Rap1-deficient naïve CD4$^+$ T cells did not show biased differentiation into Th17 cells; their differentiation into Treg cells as well as Th1 and Th2 cells was lesser than that of wild-type cells. Rap1-deficient naïve CD4$^+$ T cells were found to exhibit the defective nuclear translocation of NFAT and formation of actin foci in response to TCR engagement. These data suggest that Rap1 amplifies the TCR signaling required for Treg-mediated control of intestinal colitogenic Th17 responses.

[1] Department of Biosciences, School of Science, Kitasato University, 1-15-1 Kitasato, Minamiku, Sagamihara, Kanagawa 252-0344, Japan. [2] Immunology Discovery Research, Lilly Research Laboratories, Lilly Biotechnology Center, Eli Lilly and Company, 10290 Campus Point Drive, San Diego, CA 92121, USA. [3] Department of Immunoregulation, Institute of Medical Science, Tokyo Medical University, 6-1-1 Shinjuku, Shinjuku-ku, Tokyo 160-8402, Japan. [4] Institute for Advanced Biosciences, Keio University, 246-2 Mizukami, Kakuganji, Tsuruoka, Yamagata 997-0052, Japan. [5] Intestinal Microbiota Project, Kanagawa Institute of Industrial Science and Technology, 3-25-13 Tonomachi, Kawasaki-ku, Kawasaki, Kanagawa 210-0821, Japan. [6] Transborder Medical Research Center, University of Tsukuba, 1-1-1 Tennodai, Tsukuba, Ibaraki 305-8575, Japan. ✉email: katagirk@kitasato-u.ac.jp

The small GTPase Ras-related protein1(Rap1) is a small GTPase that regulates integrin-mediated lymphocyte adhesion and migration, which are crucial for immunosurveillance[1–3]. We previously found that the homing of Rap1-deficient T cells into peripheral lymph nodes (LNs) in mice was less than one-tenth of that in wild-type (WT) T cells because of impaired integrin-dependent arrest on high endothelial cells[4]. Hence, T-cell-specific Rap1-knockout (Rap1KO) mice show homeostatic proliferation after lymphopenia and develop spontaneous colitis with tumors[4]. However, the mechanisms by which Rap1 deficiency in T cells causes the failure of immune tolerance to microbiota remain unclear.

Regulatory T (Treg) cells suppress excess immunity against self-antigen and commensal bacteria-derived antigens[5,6]. These cells are characterized by the expression of the transcription factor Forkhead box P3 (Foxp3), which plays crucial roles in the differentiation, maintenance, and function of Treg cells[7,8]. Treg cells developed in the thymus are called thymus-derived Treg (tTreg) cells and express the transcription factor Helios[9,10]. Treg cells undergo further differentiation into fully suppressive effector Treg cells in response to T cell antigen receptor (TCR) and cytokine signals[11,12].

Peripheral Treg (pTreg) cells arise from naïve CD4$^+$Foxp3$^-$ conventional T cells (CD4$^+$ cells) at the periphery upon antigen stimulation with an appropriate combination of cytokines such as IL-2 and TGFβ[13,14]. pTreg are mostly present in the intestine, primarily because of the abundant expression of TGFβ and retinoic acid, and are thought to be particularly important in the establishment of tolerance to commensal bacteria at mucosal sites[15–17]. Most pTreg in the intestine co-express Foxp3 and RORγt. RORγt-expressing Treg cells differentiate in a TCR-dependent manner in response to intestinal microbiota. They represent a stable Treg lineage with a highly suppressive phenotype, and confer protection from intestinal immunopathology[18–20]. RORγt$^+$ Treg cell deficiency exacerbates colitogenic Th17 cell differentiation and promotes colonic inflammation[21].

Cytotoxic T-lymphocyte-associated antigen-4 (CTLA-4) plays an important role in the negative regulation of the immune system, and it prevents aberrant T-cell responses against self-proteins[22–24]. The CTLA-4 pathways represent the core mechanism of Treg cell suppression that is indispensable for normal immune homeostasis[25]. CTLA-4 on Treg cells functions in an extrinsic manner to control the CD28-dependent activation of naïve T cells by restricting their access to costimulatory ligands[26,27]. CTLA-4 is a highly endocytic molecule that captures the costimulatory ligands, CD86 and CD80 from opposing cells through trans-endocytosis[28–30]. By restricting the expression of costimulatory ligands in this manner, CTLA-4 inhibits the CD28-dependent activation of T cells.

Importantly, TCR signaling plays central roles in Treg cell differentiation, Foxp3-mediated gene regulation and suppressor functions[5,11,12,31]. Differentiating Treg cells recognize their cognate antigens, receive TCR signals, and initiate Foxp3 transcription[32]. A reporter mouse for TCR signal strength reveals that Treg cells continuously receive stronger TCR signals than conventional T cells[33]. Inhibition of the linker for activation of T cells (LAT)-PLCγ association downstream of the TCR severely interferes Treg cell, but not conventional T cell development[34]. These papers suggest that Treg cell development, maintenance and function require discrete TCR-mediated signaling. The transcription factor, nuclear factor of activated T-cells (NFAT), a key regulator of T cell activation, is involved in the differentiation and function of Treg cells by forming cooperative complex with Foxp3[35]. The interaction of NFAT with Foxp3 is necessary for the upregulation of Treg cell markers, such as CTLA-4 and IL-10, and for conferring the suppressive function to Treg cells. Ca$^{2+}$- influx via Ca$^{2+}$ release-activated Ca$^{2+}$ (CRAC) channels formed by stromal interaction molecule (STIM) and ORAI proteins is required not only for the development of Treg cells, but also for their differentiation into effector Treg cells and their suppressive function[36].

TCR ligands, including agonist anti-CD3 antibody, trigger the formation of signaling microclusters (MCs) and cause a series of changes in T cell synaptic cytoskeleton[37–42]. The disruption of global F-actin causes defects in early signaling events such as MC formation and early TCR signaling, and in distal TCR signaling events including intracellular Ca$^{2+}$ ([Ca$^{2+}$]i) increase and store-operated Ca$^{2+}$ entry[43,44]. On the other hand, the deficiency of an actin effector protein, such as the Wiscott Aldrich Syndrome Protein (WASP), which is a hematopoietic-cell-specific nucleation promoting factors, does not affect early TCR signaling and the formation of supramolecular activation clusters (SMAC) but impairs the formation of the F-actin microstructure, visualized as F-actin foci, and distal TCR signaling events[45–47].

In the present study, we found that Rap1 is involved in the TCR-dependent proliferation and differentiation of CD4$^+$ cells in an integrin-independent manner. We propose that Rap1 contributes to the maintenance of immune tolerance to microbiota in multiple aspects and prevents the development of spontaneous colitis.

## Results

**Exacerbated Th17 cell differentiation causes spontaneous colitis in T cell-specific Rap1KO mice.** We previously found that T cell-specific Rap1KO mice (Rap1KO mice), which deleted Rap1a/b in T cells by crossing *Rap1a*$^{f/f}$ and *Rap1b*$^{f/f}$ mice with CD4-cre mice, spontaneously developed severe colitis with tubular adenoma[4]. The proportion and numbers of IL-17A-expressing cells and IL-17A and IFNγ-co-expressing cells in the large intestinal lamina propria (LILP) of Rap1KO mice were fourfold or more higher than those in the LILP of WT mice (Fig. 1a). IL-17A$^+$ cells and IL-17A$^+$IFNγ$^+$ cells also increased in the mesenteric lymph nodes (mLN) and blood of Rap1KO mice (Fig. 1b). Almost half of IL-17A$^+$ cells co-expressed IL-17F (Fig. 1c). To clarify the role of Th17 cells in the development of colitis, we crossed Rap1KO mice with IL-17A KO mice (double knockout mice; DKO). IL-17A-expressing cells were absent and IL-17F-expressing cells were remaining in the LILP of DKO mice (Fig. 1c). The inhibition of IL-17A production in the DKO mice prevented the development of colitis (loss of body weight, and shorten LILP length, and epithelial hyperplasia) in Rap1KO mice (Fig. 1d). These results indicate that excess IL-17A production causes colitis in Rap1KO mice.

The administration of antibiotics inhibited the IL-17A, IL-17F and IFNγ-expressing cells in the LILP of both WT and Rap1KO mice, and prevented the development of colitis (Supplementary Fig. 1a, b), indicating that colitis is developed in microbiota-dependent manner in Rap1KO mice. Therefore, we examined the diversity of the microbial communities in the cecum contents of WT, IL-17AKO, Rap1KO and DKO mice by 16S rRNA gene sequencing. There was no difference in total microbial counts between each mice. Comparisons of gut microbiome profiles based on weighted UniFrac distances indicated that the microbiota of WT and Rap1KO mice, but not DKO mice were significantly different (Supplementary Fig. 1c). The unweighted and weighted UniFrac PCoA plots visually confirmed the distinct separating of microbiota communities between WT and Rap1KO, but not between WT and DKO mice (Supplementary Fig. 1c). Comparison of the relative abundance of the gut microbiota compositions between WT, IL-17AKO, Rap1KO and DKO mice at the family showed that there were significantly increased or decreased families in Rap1KO mice (Supplementary Fig. 1c). Thus, these data suggest that dysbiosis is caused in an inflammation-dependent manner in Rap1KO mice.

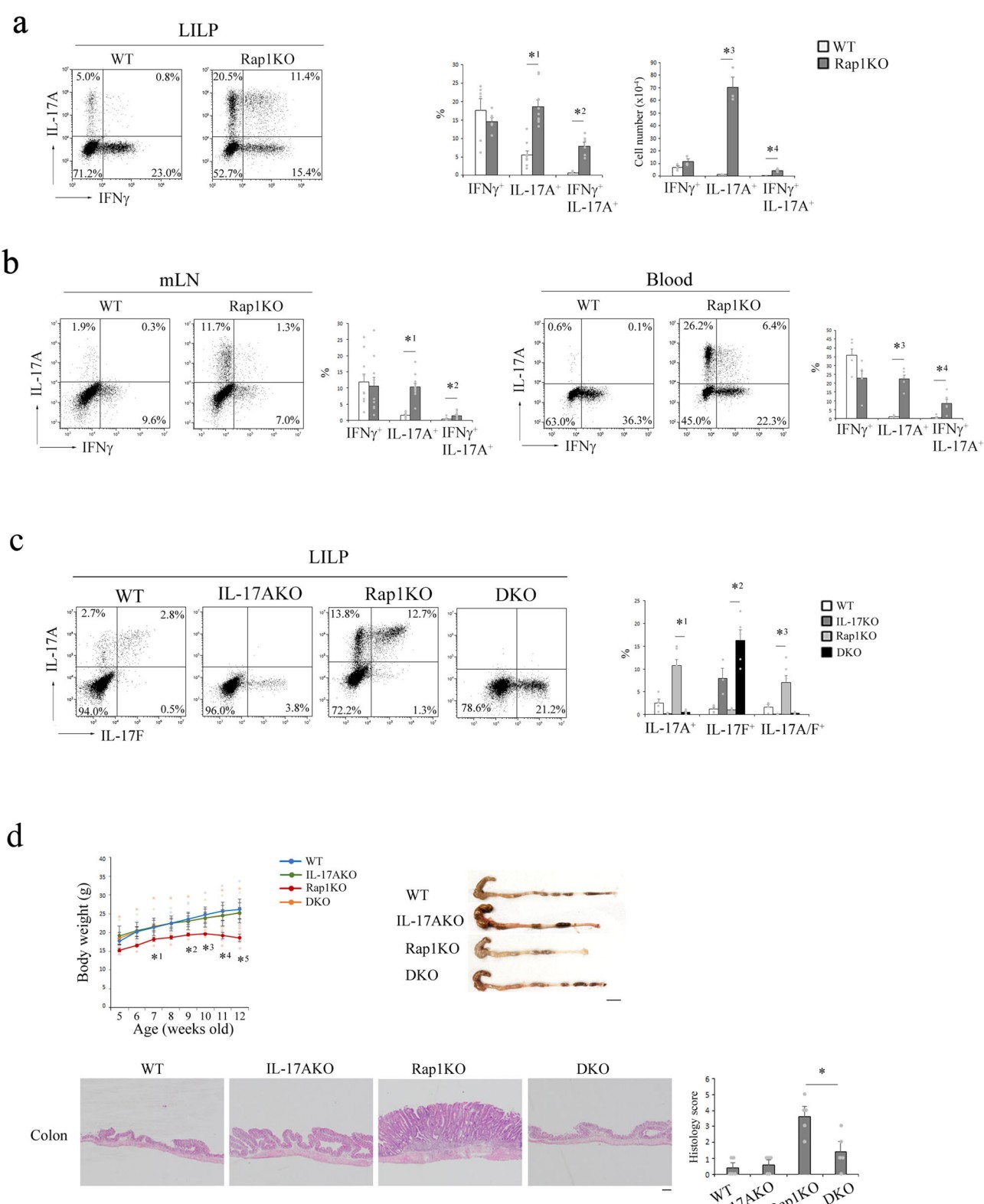

**The generation of RORγt⁺ Treg cells is impaired in the LILP of Rap1KO mice**. As RORγt⁺Foxp3⁺ (Treg) cells were critical for the inhibition of intestinal inflammation[18–20], we examined the proportions of RORγt⁺ Treg cells in the LILP of WT and Rap1KO mice from 2 weeks after birth. RORγt⁺ Treg cells were clearly detected in the LILP of WT mice from 4 weeks of age; their proportion increased with age. However, the ratio and number of RORγt⁺ Treg cells in the LILP of Rap1KO mice significantly decreased compared with those in the LILP of WT mice from 4 to 11 weeks of age (Fig. 2a and Supplementary Fig. 2a). On the other hand, the ratio and number of RORγt⁺Foxp3⁻ (Th17) cells significantly increased in the LILP of Rap1KO mice from 4 to 11 weeks of age (Fig. 2a and Supplementary Fig. 2a).

**Fig. 1 The development of colitis in Rap1KO mice was dependent on IL-17A. a** (Left) Representative IL-17A and IFNγ profiles of CD4$^+$ cells from the LILP of WT and Rap1KO mice at 8–12 weeks of age ($n = 6$–9). T cells from the LILP were stimulated for 4 h with PMA plus ionomycin. Following this, flow cytometry was performed to determine the frequency of Th1 and Th17 cells among CD4$^+$ cells based on the production of IFNγ and IL-17A, respectively. (Right) Graphs represent the mean ± S.E.M. of the ratios (left) ($n = 6$–9) to CD4$^+$ cells and numbers (right) ($n = 3$) of IFNγ-expressing, IL-17A-expressing, and IFNγ + IL-17A-co-expressing cells in the LILP. *$^1P$ < 0.001, *$^2P$ < 0.001, *$^3P$ < 0.001 and *$^4P$ < 0.009 compared with WT mice. **b** Representative IL-17A and IFNγ profiles of CD4$^+$ cells from the mesenteric lymph nodes (mLNs) (left) or blood (right) of WT and Rap1KO mice at 8 weeks of age ($n = 6$–10). Graphs represent the mean ± S.E.M. of the ratios of IFNγ-expressing, IL-17A-expressing, and IFNγ + IL-17A-co-expressing cells to CD4$^+$ cells in the mLN and blood, respectively. *$^1P$ < 0.001, *$^2P$ < 0.006, *$^3P$ < 0.001 and *$^4P$ < 0.007 compared with WT mice. **c** (Left) Representative IL-17A and IL-17F profiles of CD4$^+$ cells from the LILP of WT, IL-17AKO, Rap1KO and DKO mice ($n = 3$–6). (Right) Graph represents the mean ± S.E.M. of the ratios of IL-17A-expressing, IL-17F-expressing, and IL-17A + IL-17F-co-expressing cells to CD4$^+$ cells in the LILP. *$^1P$ < 0.001, *$^2P$ < 0.001, and *$^3P$ < 0.02 compared with WT mice. **d** (Upper left) The body weights of WT, IL-17AKO, Rap1KO, and DKO mice ($n = 3$–6) were measured every week. Data represent the mean ± S.E.M. *$^1P$ < 0.05, *$^2P$ < 0.05, *$^3P$ < 0.04, *$^4P$ < 0.02, and *$^5P$ < 0.007 compared with WT mice. (Upper right) Representative LI morphology of WT, IL-17AKO, Rap1KO, and DKO mice at 12 weeks of age. Scale bar, 1 cm. (Lower left) Representative histology of intestinal inflammation. Paraffin-embedded LI sections from WT, IL-17AKO, Rap1KO, and DKO mice were stained with hematoxylin and eosin. Representative low (40×) magnification histological images of LI tissues are shown. Scale bars, 200 μm. (Lower right) Light microscopic assessment of colitis damage in WT, IL-17AKO, Rap1KO, and DKO mice, as described in the Methods. Data represent the mean ± S.E.M. ($n = 5$). *$P$ < 0.007 compared with Rap1KO mice.

The appearance of RORγt$^+$ Treg cells was inhibited by the administration of antibiotics in both WT and Rap1KO mice (Fig. 2a). A reduction in the proportion of RORγt$^+$ Treg cells was similarly observed in the LILP of DKO and Rap1KO mice (Fig. 2b), suggesting that the impaired generation of RORγt$^+$ Treg cells in the LILP was not caused by inflammation.

The proportion of RORγt$^-$ Treg cells, which were Helios$^+$, slightly decreased until 4 weeks of age but rather increased at 11 weeks of age after the onset of colitis (Fig. 2a). On the other hand, the expression levels of Foxp3 and Helios in Treg cells in the thymus of Rap1KO mice were significantly lower than those in the thymus of WT mice (Supplementary Fig. 2b), and the proportion of Treg cells among CD4$^+$ cells in the spleen of Rap1KO mice was significantly reduced at 4 weeks of age (Supplementary Fig. 2c), indicating that Treg cell generation was not optimal in the thymus of Rap1KO mice. However, the proportions of Treg cells among CD4$^+$ cells in the mLNs of Rap1KO mice were not significantly diminished at 4 and 7 weeks of age (Supplementary Fig. 2d).

As RORγt$^+$ Treg cells are reported to mainly differentiate from adult naïve CD4$^+$ cells in LILP[14], we transferred WT CD4$^+$CD25$^-$ cells isolated from adult congenic mice (Ly5.1$^+$) into 6-wk-old Rap1KO mice (Ly5.2$^+$)(Fig. 3a). Four weeks later, the injected WT CD4$^+$ cells (Ly5.1$^+$) showed the dominant CD4$^+$ population (more than 70%) in the LILP of the recipient Rap1KO mice (Ly5.2$^+$) (Fig. 3b). As previously reported[14], WT CD4$^+$CD25$^-$ cells (Ly5.1$^+$) differentiated into RORγt$^+$ Treg cells in the LILP of Rap1KO mice (Fig. 3b). The proportion of RORγt$^+$ cells in the WT CD4$^+$Foxp3$^+$ cells was significantly higher than that of the recipient Rap1KO mice, although WT naïve CD4$^+$CD25$^-$ cells also differentiated into Th17 cells (Fig. 3c). The proportions of IL-17A$^+$ and IL-17A$^+$ IFNγ$^+$ cells were significantly reduced in both WT and Rap1-deficient CD4$^+$ cells of the recipient mice, compared with those of control non-transferred Rap1KO mice (Fig. 3b, c). The development of colitis (loss of body weight, and epithelial hyperplasia) in the recipient Rap1KO mice was significantly reduced compared with that of non-transferred Rap1KO mice (Fig. 3d). These data suggest the reduced capacity of Rap1-deficient naïve CD4$^+$ cells to differentiate RORγt$^+$Treg cells might in part lead to the development of pathogenic Th17 cells and colitis in Rap1KO mice.

**Rap1-deficiency impairs the TCR-dependent induction of CTLA-4 and IL-10 production.** CTLA-4 plays essential roles in the suppressive function of Treg cells[28]. In addition, the inhibition of CD4$^+$ T cell expansion by Treg cells in a lymphopenic environment requires CTLA-4[48,49]. The expression of CTLA-4 was significantly lesser on both RORγt$^+$ and RORγt$^-$ Treg cells in the LILP and

mLNs of Rap1KO mice than in those of WT mice (Fig. 4a). These results suggest that CTLA-4-dependent trans-endocytosis of costimulatory ligands is reduced in Rap1KO mice. Therefore, we examined the expression of CD80 and CD86 on dendritic cells (DCs) in the mLNs and LILP of Rap1KO mice. The presentation of tissue-derived self-antigens could trigger Treg cells to capture costimulatory ligands in vivo, and migratory DCs are targets for Treg-based CTLA-4-dependent regulation in the steady state[50]. As shown in Fig. 4b, the expression of CD80 and CD86 on both CD11c$^+$MHCII-$^{high}$CD103$^+$CD11b$^+$ and CD11b$^-$ migratory DCs (Supplementary Fig. 3a) was significantly higher in the mLNs of Rap1KO mice than in those of WT mice. The expression of CD80 and CD86 on CD11c$^+$ MHC II$^{high}$CD103$^+$CD11b$^-$ and CD103$^-$CD11b$^+$ DCs, which were the main subsets of DCs in the LILP (Supplementary Fig. 3a), was also significantly higher in the LILP of Rap1KO mice than in the LILP of WT mice (Fig. 4b).

As Treg cell-derived IL-10 is reported to limit the inflammation at the colon[51], we also examined the expression of IL-10 in the RORγt$^+$ and RORγt$^-$ Treg in the LILP of Rap1KO mice. As shown in Fig. 4c, the expression of IL-10 was significantly lesser in both RORγt$^+$ and RORγt$^-$ Treg cells in the LILP of Rap1KO mice than in those of WT mice.

Therefore, we assessed the effects of Rap1 deficiency on the suppressive role of Treg cells in the anti-CD3-stimulated proliferation of WT naïve CD4$^+$ T cells (responder cells) using bone marrow-derived dendritic cells (BMDCs) as costimulatory ligand (CD80 and CD86)-expressing cells instead of anti-CD28. Anti-CD3-stimulated responder cell proliferation was observed only in the presence of BMDCs. Treg cells were isolated from the spleens of WT and Rap1KO mice as CD4$^+$CD25$^+$ cells (Supplementary Fig. 3b). The addition of WT Treg cells suppressed the BMDC-dependent proliferation of responder cells; however, the suppressive capacity of Rap1-deficient Treg cells was significantly lesser than that of WT Treg cells (Fig. 4d). There was no difference in proliferative responses to anti-CD3 and CD28 between WT or Rap1-deficeint Treg cells (Supplementary Fig. 3b). Those data suggest a defect in the suppressive functions of Rap1-deficient Treg cells.

**Rap1 deficiency affects in vitro differentiation of naïve CD4$^+$ T cells into effector and regulatory subsets.** To determine whether the differentiation of Rap1-deficient naïve CD4$^+$ cells was biased toward Th17 cells, the cells were stimulated by crosslinking with low (0.05 μg/ml) or high (0.5 μg/ml) concentrations of anti-CD3 in the presence of anti-CD28 and cultured under each polarizing condition. Dead cells were detected using zombie dye and excluded from the analysis of expression of

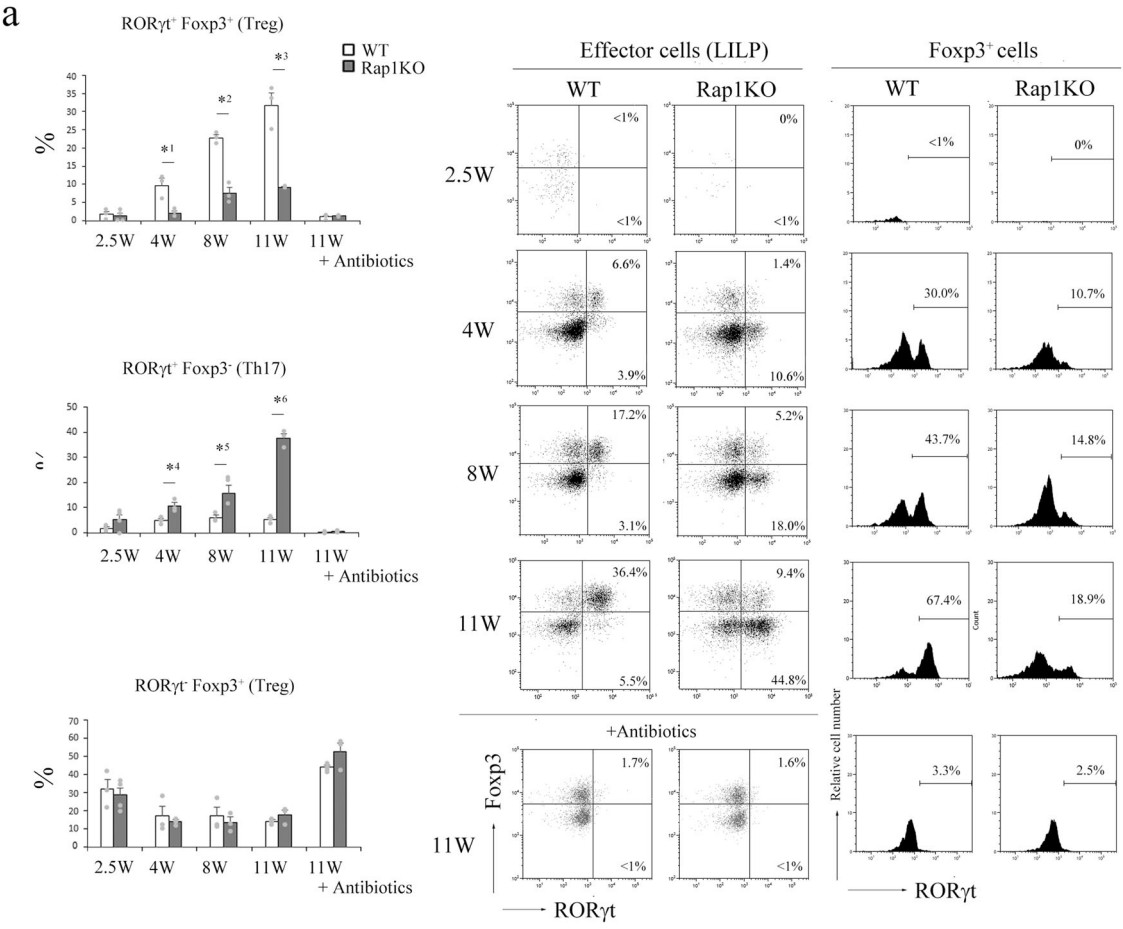

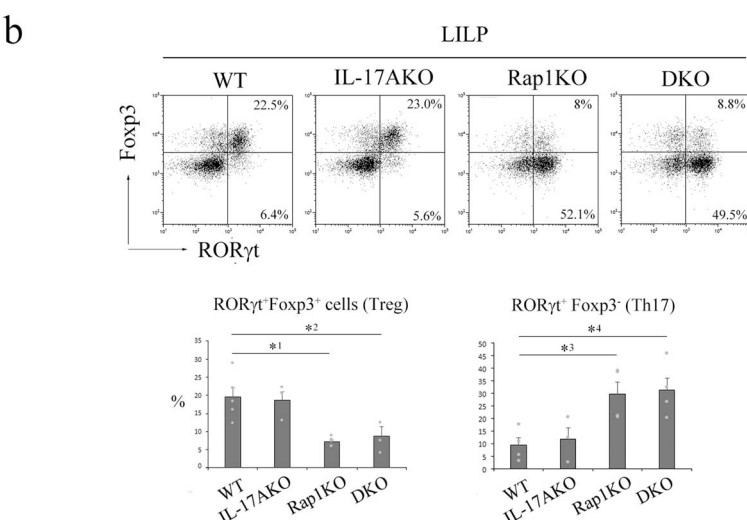

**Fig. 2 The percentages of RORγt⁺ Treg cells decreased in the LILP of Rap1KO mice. a** (Left) The percentages of RORγt⁺Foxp3⁺ cells (top), RORγt⁺Foxp3⁻ cells (middle), and RORγt⁻Foxp3⁺ cells (bottom) to effector CD4⁺ cells in the LILP of WT and Rap1KO mice with or without the administration of antibiotics at 2.5–11 weeks of age ($n = 3$–4). Data represent the mean ± S.E.M. *[1]$P < 0.02$, *[2]$P < 0.001$, *[3]$P < 0.003$, *[4]$P < 0.03$, *[5]$P < 0.03$, and *[6]$P < 0.001$ compared with the corresponding WT mice. (Right) Representative RORγt and Foxp3 profiles of effector CD4⁺ cells (left) and RORγt histogram profiles of Foxp3⁺ cells (right) from the LILP of WT and Rap1KO mice at the indicated age (weeks). **b** (Upper) Representative RORγt and Foxp3 profiles of effector CD4⁺ cells from the LILP of WT, IL-17AKO, Rap1KO, and DKO mice at 8–12 weeks of age. (Lower) The percentages of RORγt⁺Foxp3⁺ cells (left) and RORγt⁺Foxp3⁻ cells (right) to CD4⁺ cells derived from the LILP of WT, IL-17AKO, Rap1KO, and DKO mice at 8–12 weeks of age ($n = 3$–5). Data represent the mean ± S.E.M. *[1]$P < 0.005$, *[2]$P < 0.03$, *[3]$P < 0.02$ and *[4]$P < 0.02$ compared with the corresponding WT mice.

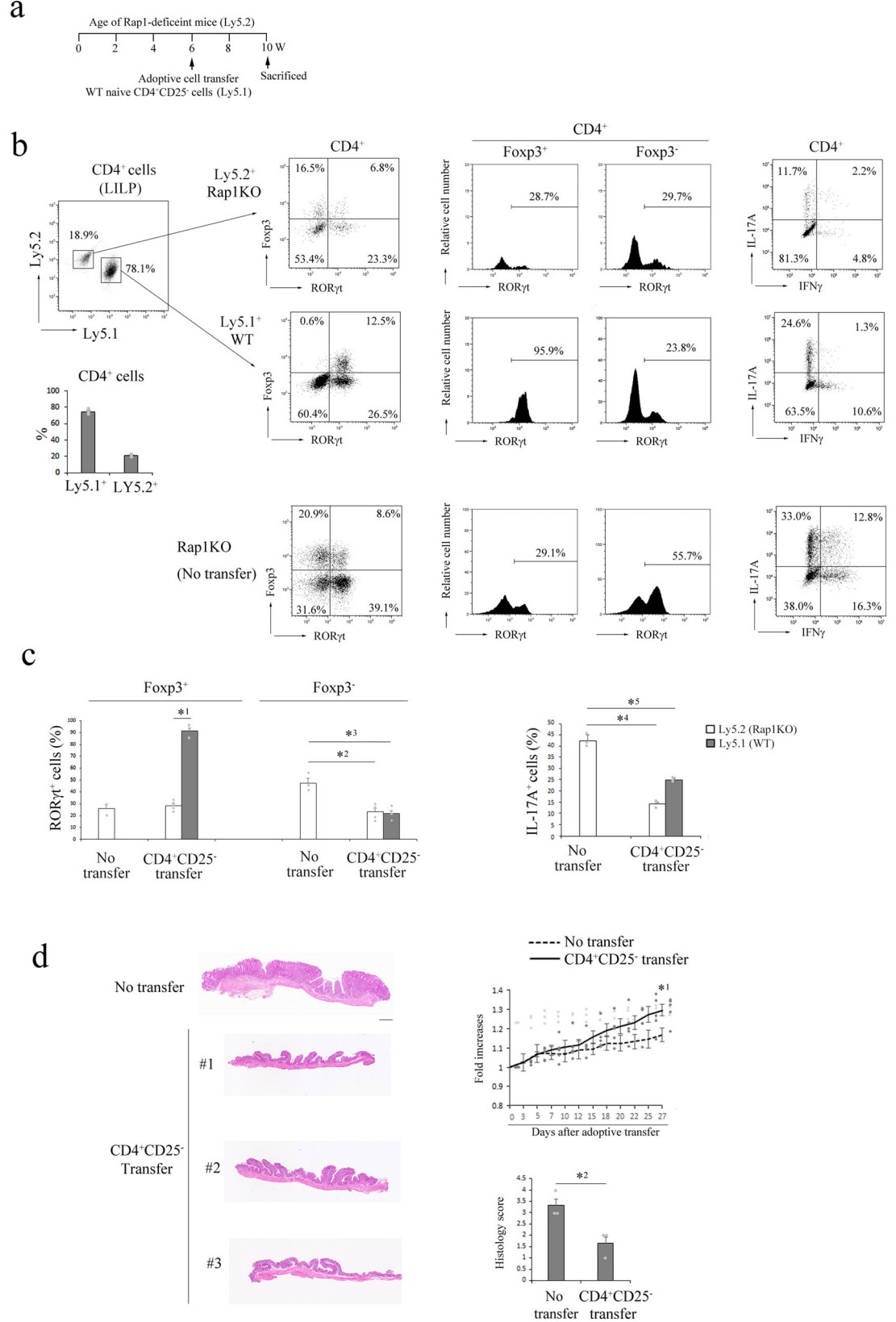

transcription factor and cytokines. As shown in Fig. 5a, RORγt expression in Rap1-deficient naïve CD4+ cells under Th17-polarizing conditions was comparable to that in WT naïve CD4+ cells, while IL-17 production in Rap1-deficient naïve CD4+ cells was lesser than that in WT naïve CD4+ cells in the presence of

both concentrations of anti-CD3 (Supplementary Fig. 4a). This finding indicates that Rap1-deficient naïve CD4+ cells do not show Th17 bias.

Under Treg-polarizing conditions, WT naïve CD4+ cells mostly differentiated into Treg cells in the presence of a low

**Fig. 3 Adoptive transfer of naïve CD4$^+$ cells prevented the development of colitis in Rap1KO mice. a** Schematic diagram of adoptive cell transfer. **b** (Left) Representative Ly5.1 and Ly5.2 profiles of CD4$^+$ cells from the LILP of recipient Rap1KO Ly5.2$^+$ mice which were transferred with WT naïve CD4$^+$CD25$^-$ cells isolated from the donor Ly5.1$^+$ mice (upper). Graph shows the proportions of Ly5.1$^+$ (donor) and Ly5.2$^+$ (recipient) cells among CD4$^+$ cells ($n = 3$) (lower). (Right) Representative Foxp3 and RORγt profiles of Ly5.2$^+$ or Ly5.1$^+$ CD4$^+$ cells (left), RORγt histogram profiles of Foxp3$^+$ or Foxp3$^-$ cells among Ly5.2$^+$ or Ly5.1$^+$ CD4$^+$ cells (center), and IL-17A and IFNγ profiles of Ly5.2$^+$ or Ly5.1$^+$ CD4$^+$ cells (right) from the LILP of the transferred or non-transferred Rap1KO mice. **c** (Left) The proportions of RORγt$^+$ cells to Foxp3$^+$ and Foxp3$^-$ cells in Ly5.2$^+$ or Ly5.1$^+$ CD4$^+$ cells from the LILP of the transferred or non-transferred Rap1KO mice ($n = 3$–5). (Right) The proportions of IL-17A$^+$ cells to Ly5.2$^+$ or Ly5.1$^+$ CD4$^+$ cells from the LILP of the transferred or non-transferred Rap1KO mice. Data represent the mean ± S.E.M. *$^1$$P < 0.001$ compared with transferred Rap1KO mice (Ly5.2$^+$), *$^2$$P < 0.003$, *$^3$$P < 0.001$, *$^4$$P < 0.001$ and *$^5$$P < 0.001$ compared with control mice. **d** (Left) Representative histology of intestinal inflammation. Paraffin-embedded LI sections from three Rap1KO mice which were transferred with WT naïve CD4$^+$CD25$^-$ T cells, or non-transferred Rap1KO mice, were stained with hematoxylin and eosin. Representative low (40×) magnification histological images of LI tissues are shown. (Right upper) The body weights of transferred and non-transferred Rap1KO mice were measured every 2 or 3 days and are presented as the fold increases relative to the original body weights ($n = 3$–5). Data represent the mean ± S.E.M. *$^1$$P < 0.04$, compared with the non-transferred Rap1KO mice. (Right lower) Light microscopic assessment of colitis damage in transferred and non-transferred mice. Data represent the mean ± S.E.M. ($n = 3$). *$^2$$P < 0.03$ compared with the non-transferred Rap1KO mice. Scale bar, 500 μm.

concentration of anti-CD3, and the expression of Foxp3 was significantly reduced in Rap1-deficient naive CD4$^+$ cells compared with that of WT cells (Fig. 5b); there was no difference in apoptosis and proliferation between WT and Rap1-deficient naïve CD4$^+$ cells in this condition (Supplementary Fig. 4b). However, a high concentration of anti-CD3 induced apoptosis and reduced the proliferation of WT naïve CD4$^+$ cells (Supplementary Fig. 4b). The means of Foxp3 expression were decreased and not different in WT and Rap1-deficient cells in the presence of a high concentration of anti-CD3 (Fig. 5b). In addition, when cultured under Treg-polarizing conditions in the presence of PMA plus ionomycin, there was no difference in the expression of Foxp3 between WT and Rap1-deficient cells (Fig. 5b), suggesting that defective TCR-mediated signaling reduces Foxp3 expression in Rap1-deficient cells. These results indicate that Rap1 was required for optimal induction of Foxp3 in naïve CD4$^+$ cells.

The expression of T-bet and GATA3, and cytokine production in Rap1-deficient naïve CD4$^+$ cells cultured under Th1-polarizing conditions and Th2-polarizing conditions, respectively, were significantly lesser than those in WT naïve CD4$^+$ cells (Supplementary Fig. 4c, d). Under Th1-polarizing conditions and Th2-polarizing conditions, apoptosis and proliferation were not reduced in Rap1-deficient cells (Supplementary Fig. 4e).

These findings indicate that Rap1 deficiency reduces the differentiation of naïve CD4$^+$ cells into regulatory and effector subsets in vitro.

Naïve CD4$^+$ cells are capable of expressing CTLA-4 and performing trans-endocytosis of CD80 and CD86 upon TCR stimulation[29]. We examined the effects of Rap1 deficiency on TCR-induced CTLA-4 expression and trans-endocytosis in naïve CD4$^+$ cells using CD80-GFP-expressing CHO cells. TCR-dependent CTLA-4 expression and trans-endocytosis of CD80-GFP were significantly lesser in Rap1-deficient naïve CD4$^+$ cells than in WT naïve CD4$^+$ cells (Fig. 5c). However, the expression of CTLA-4 and trans-endocytosis of CD80-GFP induced by PMA plus ionomycin stimulation did not differ between WT and Rap1-deficient naïve CD4$^+$ cells (Fig. 5c), suggesting that Rap1 deficiency is not directly involved in CTLA-4-dependent trans-endocytosis of costimulatory ligands but may impair TCR-mediated signaling.

**Rap1 is involved in TCR signaling**. We examined the proliferative responses of naïve CD4$^+$ cells derived from WT and Rap1KO mice to various concentrations (0.001–0.5 μg/ml) of anti-CD3 in the presence of anti-CD28. Rap1-deficient naïve CD4$^+$ cells did not proliferate in the presence of a low dose of anti-CD3 (0.005 μg/ml); however, in the presence of a high dose

of anti-CD3 (0.5 μg/ml), they proliferated more than WT naïve CD4$^+$ cells at 48 h after stimulation (Fig. 6a). Thus, Rap1-deficient naïve CD4$^+$ cells needed a higher concentration of anti-CD3 for proliferation than WT CD4$^+$ cells (Fig. 6a, b). On the other hand, cell death was induced in WT naïve CD4$^+$ cells but not in Rap1-deficient naïve CD4$^+$ cells at 48 h after stimulation with more than 0.05 μg/ml of anti-CD3 (Fig. 6c). The tyrosine phosphorylation of ZAP70, SLP76, and PLCγ, the phosphorylation of extracellular regulated-kinase (ERK) and c-Jun; and expression of myc in Rap1-deficient CD4$^+$ cells were similar to those in WT naïve CD4$^+$ cells after stimulation with anti-CD3 and anti-CD28 (Fig. 6d and Supplementary Fig. 7). As shown in Fig. 6e, Ca$^{2+}$ influx, subsequent STIM-1 translocation, and Ca$^{2+}$ influx-dependent cell spreading were impaired in Rap1-deficient naïve CD4$^+$ cells after the stimulation with anti-CD3 and anti-CD28. Similarly, the nuclear translocation of NFAT1 was inhibited in Rap1-deficient naïve CD4$^+$ cells after the stimulation with anti-CD3 and anti-CD28 (Fig. 6e).

**Rap1 is critical for the formation of actin foci**. As the reorganization of the actin cytoskeleton is required for Ca$^{2+}$ entry following TCR engagement[45], we examined the effects of Rap1 deficiency on actin reorganization in response to TCR engagement. WT naïve CD4$^+$ cells demonstrated clearly punctate F-actin microstructures, whereas Rap1-deficient naïve CD4$^+$ cells did not show these microstructures after incubation for 10 min on a surface coated with anti-CD3 and anti-CD28 (Fig. 7a). Therefore, we examined whether the punctate F-actin microstructures were actin foci, which were made of branched filaments driven by WASP and maintained by phosphorylated HS1[40,45]. As shown in Fig. 7b, the punctate F-actin microstructures co-localized with WASP and phosphorylated HS1 in WT naive CD4$^+$ cells; however, these were not observed in Rap1-deficient naïve CD4$^+$ cells. A previous study revealed an increase in the phosphorylation levels of HS1 with the formation of actin foci[47]. The phosphorylation of HS1 was not augmented in Rap1-deficient naive CD4$^+$ cells at 30 min after stimulation (Fig. 7b and Supplementary Fig. 7). Signaling MCs containing phosphorylated ZAP70 (pZAP70) and phosphorylated SLP76 (pSLP76) were partially co-localized with actin foci in WT naïve CD4$^+$ cells (Fig. 7c). In Rap1-deficient naïve CD4$^+$ cells, MCs containing pZAP70 and pSLP76 were normally formed; however, F-actin was not developed at the MC sites (Fig. 7c).

Furthermore, we examined the Rap1-GTP generation sites using Ral-guanine nucleotide dissociation inhibitor (GDS)-Ras-binding domain (RBD)-mCherry as a reporter, and Lifeact-GFP as a F-actin reporter. Ral-GDS-RBD-mCherry showed the punctate microstructures, which were partially co-localized with actin foci in response

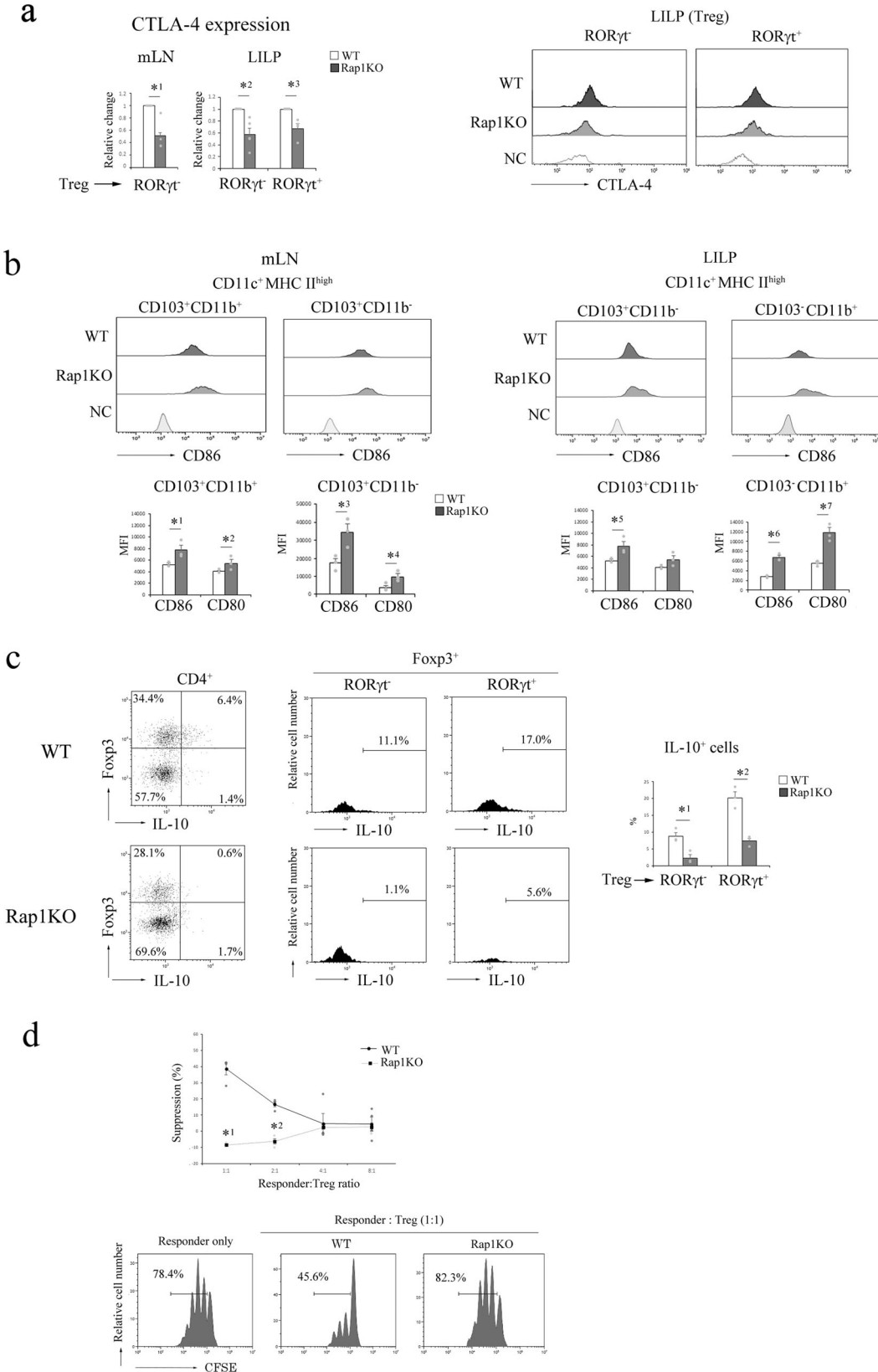

to TCR engagement (Fig. 7d and Supplementary Fig. 5a). Ral-GDS-RBD-mCherry also partially co-localized with WASP-GFP (Fig. 7d and Supplementary Fig. 5a).

Previous studies demonstrated that RIAM (Rap1-GTP-interacting adaptor molecule), actin regulator[52], was involved in TCR-mediated signaling. We examined whether RIAM and lamellipo-din (Lpd), which is another member of the MRL (Mig-10/RIAM/Lpd) family of proteins, were co-localized with actin foci in WT naïve CD4+ cells. As shown in Supplementary Fig. 5b, both RIAM and Lpd were present at the membrane ruffles, but were

**Fig. 4 Rap1 deficiency impaired the functions of Treg cells. a** (Left) The expression of CTLA-4 on RORγt⁻Foxp3⁺ and RORγt⁺Foxp3⁺ T cells in the mLNs and LILP of WT and Rap1KO mice (n = 4–5) at 7 weeks of age. The expression of CTLA-4 on WT and Rap1-deficient Treg cells is presented as the fold change in the normalized MFI (median fluorescence intensity) of CTLA-4 on Rap1-deficient CD4⁺ cells relative to that on WT CD4⁺ cells (adjust 1). Graphs represent the mean ± S.E.M. *¹P < 0.009, *²P < 0.004, and *³P < 0.009 compared with WT mice. (Right) Representative flow cytometry profiles of CTLA-4 and the negative control (NC) expressed on RORγt⁻Foxp3⁺ or RORγt⁺Foxp3⁺ T cells in the LILP of WT and Rap1KO mice. **b** (Left upper) Representative flow cytometry profiles of CD86 expressed on CD11c⁺MHCII^high CD103⁺CD11b⁺ and CD11b⁻ DCs in the mLNs of WT and Rap1KO mice. (Left lower) MFIs of CD86 and CD80 expressed on CD11c⁺MHCII^high CD103⁺CD11b⁺ and CD11b⁻ DCs in the mLNs of WT and Rap1KO mice (n = 3) at 7 weeks of age. Data represent the mean ± S.E.M. *¹P < 0.02, *²P < 0.04, *³P < 0.02, and *⁴P < 0.05 compared with WT mice. (Right upper) Representative flow cytometry profiles of CD86 expressed on CD11c⁺MHCII^high CD103⁺CD11b⁻ and CD103⁻CD11b⁺ DCs in the LILP of WT and Rap1KO mice. (Right lower) MFIs of CD86 expressed on CD11c⁺MHCII^high CD103⁺CD11b⁻ and CD103⁻CD11b⁺ DCs in the LILP of WT and Rap1KO mice (n = 3) at 7 weeks of age. Data represent the mean ± S.E.M. *⁵P < 0.05, *⁶P < 0.001, and *⁷P < 0.005 compared with WT mice. **c** (Left) Representative IL-10 and Foxp3 profiles of CD4⁺ T cells in the LILP of WT and Rap1KO mice. (Center) IL-10 histogram profiles of RORγt⁻ or RORγt⁺ Foxp3⁺ cells in the LILP of WT and Rap1KO mice (n = 3) at 8 weeks of age. (Right) The proportion of IL-10-expressing cells in WT and Rap1-deficient RORγt⁻ or RORγt⁺ Foxp3⁺ cells. Graphs represent the mean ± S.E.M. *¹P < 0.02, *²P < 0.004 compared with WT mice. **d** (Upper) Treg-mediated suppression. CFSE-labeled WT naïve CD4⁺ cells (responder cells) were mixed at the indicated ratios with WT or Rap1-deficient Treg cells and stimulated with anti-CD3 in the presence of BMDCs, as described in the Methods (n = 4). Data represent the mean ± S.E.M. *¹P < 0.001 and *²P < 0.001 compared with WT Treg cells. (Lower) Representative histogram profiles of CFSE in the responder cells cultured with or without WT or Rap1-deficient Treg cells. The responder cells did not proliferate in the absence of BMDCs.

not co-localized with actin foci (Supplementary Fig. 5b). In addition, we deleted RIAM in 3A9 T cells using the CRISPR/Cas9 system. RIAM-knockout cells demonstrated distinct morphology, but formed actin foci in response to TCR engagement, although the formation of actin foci was reduced in Rap1-knockdown cells (Supplementary Fig. 5c and 7), suggesting that RIAM might not be involved in the formation of actin foci. In an in vitro pull-down assay using Rap1-GST, WASP was found to be associated with both GDP- and GTP-binding forms of Rap1 (Supplementary Fig. 5d), suggesting that Rap1 directly or indirectly associate with WASP and support the recruitment of WASP in actin foci.

As actin foci have been found to play an essential role in the activation of PLCγ and in Ca²⁺ entry in T cells[45], our findings suggest that Rap1 can facilitate TCR signaling by promoting the formation of actin foci.

## Discussion

As Rap1 is indispensable for integrin-dependent homing of naïve T cells into peripheral lymphoid tissues, Rap1 deficiency causes lymphopenia in mesenteric lymph nodes and LILP[4]. Previous papers[53–55] reported that lymphopenia-induced spontaneous proliferation of naïve T cells generates colitogenic Th17 cells in the absence of Treg cells in a microbiota-dependent manner. In the present study, we found that Rap1 deficiency impaired the preceding generation of RORγt⁺ Treg cells from naïve CD4⁺ T cells in the LILP by 4 weeks of age. RORγt⁺ Treg cells are predominantly induced in a microbiota-dependent manner and suppress the differentiation and expansion of pathogenic Th17 cells in the LILP[15,16,18–21]. Therefore, the reduced number of naïve CD4⁺ cells and defective differentiation into RORγt⁺ Treg cells might be reasons why Rap1KO mice develop spontaneous colitis (Supplementary Fig. 6).

Rap1-deficiency impaired not only Treg cell differentiation, but also Th1 and Th2 cell differentiation in vitro. Using PMA and ionomycin, instead of anti-CD3, there was no difference in Treg cell differentiation between WT and Rap1-deficient naïve CD4⁺ cells. These data suggest that Rap1 is not specifically involved in Treg cell differentiation, but TCR-dependent differentiation is impaired in Rap1-deficient naïve CD4⁺ cells. In addition, Foxp3 and Helios in tTreg cells were also significantly lowered in the thymus of Rap1KO mice, resulting in the reduced proportion of tTreg cells in the spleen of Rap1KO mice. TCR signals are critical for Treg cell differentiation, Foxp3-mediated gene regulation and suppressor functions[5,11,12,31]. Previous papers suggest that strong TCR

signaling, especially PLCγ and Ca²⁺ signaling, is necessary for Treg cell development, maintenance or function[33–36]. Therefore, defective TCR-mediated signals might reduce the generation of RORγt⁺ Foxp3⁺ cells, instead increase RORγt⁺Foxp3⁻ cells in the LILP of Rap1KO mice.

In the absence of RORγt⁺ Treg cells, lymphopenia induced the expansion of microbe-reactive Th17 cells, which leads to the intestinal inflammation[54]. Intestinal inflammation affects epithelial barrier functions and causes dysbiosis. Certain species of microbes were reported to accelerate IL-17A production[20,56]. The impaired barrier functions and dysbiosis might facilitate the biased differentiation into Th17 cells and cytokine production in the LILP of Rap1KO mice (Supplementary Fig. 6).

The CTLA-4 pathways are reported to represent the core mechanism of Treg cell suppression that is indispensable for immune homeostasis[25]. In addition, the inhibition of CD4⁺ cell expansion by Treg cells in a lymphopenic environment requires CTLA-4[48,49]. We found that Rap1-deficiency reduced TCR-dependent, but not PMA and ionomycin-dependent CTLA-4 induction and trans-endocytosis of CD80/86 in naïve CD4⁺ cells in vitro. These results suggest that Rap1 is not directly involved in trans-endocytosis of CD80/86, but defective TCR-signals by Rap1-deficiency impair the induction of CTLA-4. Rap1-deficiency also caused profound reduction in IL-10 production by Treg cells in the LILP. These data suggest that Rap1 plays crucial roles in the TCR-mediated functions of Treg cells.

LFA-1-ICAM-1-mediated adhesion of T cells with antigen-presenting cells (APCs) was reported to be involved in thymic development of tTreg cells and optimal pTreg differentiation in the LNs[57–59], and in the suppressor function of Treg cells[58,59]. Treg cell differentiation was reported to require strong TCR signals[33]. LFA-1-ICAM-1-mediated interaction augments TCR signals especially for low affinity[60]. The activation of LFA-1 by Rap1 signaling is essential for the interaction between naïve T cells and APCs, which promotes T cell recognition of antigen-MHC complexes present on APCs[61–63]. Mice expressing the constitutively active Rap1 mutant, Rap1E63 displayed increased the frequencies of tTreg and pTreg cells in the spleen and LNs[64]. On the other hand, they also reported that Rap1E63 promoted the generation of pTreg and tTreg cells via LFA-1-independent mechanisms[64]. Thus, Rap1 plays crucial role in the generation and functions of Treg cells through the regulation of both LFA-1- and TCR- dependent adhesion with APC in vivo.

Rap1-deficient naïve CD4⁺ cells showed lower proliferative responses to the stimulation by the crosslinking of TCR with a

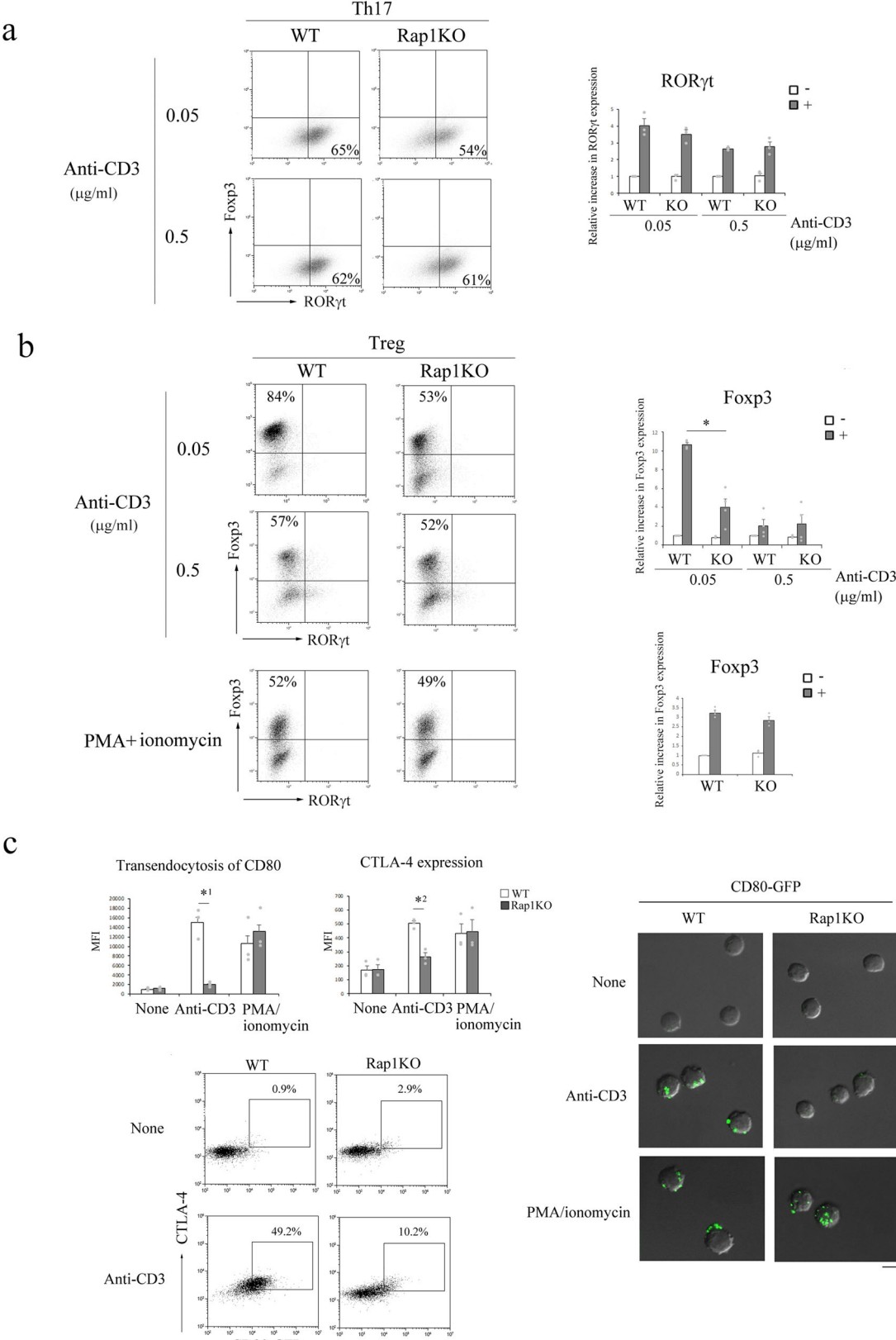

low concentration of anti-CD3 in the presence of anti-CD28 than WT naïve CD4[+] cells. T cells have been reported to reorganize the actin cytoskeleton in response to TCR engagement, which is indispensable for TCR-mediated signaling[43]. Three discrete actin networks, namely the lamellipodial branched-actin network, the lamellar actomyosin network and actin foci, have been found to be formed on the adhesion site to APC or anti-CD3-coated surface[37–42]. Actin foci are unnecessary for early TCR signaling but play an important role in distal TCR signaling events, such as $Ca^{2+}$ influx and the nuclear translocation of NFAT[45]. In the

**Fig. 5 The differentiation of Rap1-deficient naïve CD4+ cells into Treg cells was defective in vitro. a** (Left) Representative Foxp3 and RORγt profiles of naïve CD4+ cells from WT and Rap1KO mice; the cells were cultured for 3 days under Th17-polarizing conditions in the presence of different concentrations (0.05 and 0.5 μg/ml) of anti-CD3 and 2.5 μg/ml of anti-CD28, subjected to flow cytometry in order to determine the prevalence of RORγt+ cells among CD4+ cells. Data are representative of three independent experiments. (Right) The induction of RORγt in WT and Rap1-deficient naïve CD4+ cells is presented as the fold increase in the normalized MFI of RORγt in the cells cultured under Th17-polarizing conditions (+) relative to that on WT naïve CD4+ cells cultured with anti-CD3 and anti-CD28 (−) (adjust 1). Graphs represent the mean ± S.E.M (n = 3). **b** (Left) Representative Foxp3 and RORγt profiles of naïve CD4+ cells from WT and Rap1KO mice; the cells were cultured for 3 days under Treg-polarizing conditions in the presence of 0.05 or 0.5 μg/ml of anti-CD3 and anti-CD28 (upper), or PMA plus ionomycin (lower) and subjected to flow cytometry in order to determine the prevalence of Foxp3+ cells among CD4+ cells. Data are representative of three independent experiments. (Right) The induction of Foxp3 in WT and Rap1-deficient naïve CD4+ cells is presented as the fold increase in the normalized MFI of Foxp3 in the cells cultured under Treg-polarizing conditions (+) relative to that on WT naive CD4+ cells cultured with anti-CD3 and anti-CD28, or PMA plus ionomycin (−) (adjust 1). Graphs represent the mean ± S.E.M (n = 3-4). *P < 0.001 compared with the corresponding WT naïve CD4+ cells. **c** (Left upper) (left) MFI of CD80-GFP expressed on WT and Rap1-deficient naïve CD4+ cells stimulated with or without anti-CD3 or PMA plus ionomycin for 24 h (n = 3). Data represent the mean ± S.E.M. *1P < 0.003 compared with the corresponding WT naïve CD4+ cells. (right) MFI of CTLA-4 expressed on WT and Rap1-deficient naïve CD4+ cells stimulated with or without anti-CD3 or PMA plus ionomycin for 24 h (n = 3). Data represent the mean ± S.E.M. *2P < 0.003 compared with the corresponding WT naïve CD4+ cells. (Left lower) Representative flow cytometry profiles of CTLA-4 and CD80-GFP expressed on WT and Rap1-deficient naive CD4+ cells stimulated with or without anti-CD3. (Right) Representative confocal images of CD80-GFP expressed on WT and Rap1-deficient naïve CD4+ cells stimulated without (none) or with anti-CD3 and PMA plus ionomycin. Scale bar, 5 μm.

present study, Rap1-deficient naive CD4+ cells did not show defects in early TCR-mediated tyrosine phosphorylation but exhibited impairment in distal TCR signaling events such as Ca$^{2+}$ signaling. Although ZAP70 and SLP76 MCs were normally formed, actin foci were not developed in Rap1-deficient naïve CD4+ cells stimulated with anti-CD3 and anti-CD28, suggesting the involvement of Rap1 in TCR-dependent actin organization.

Actin foci are made of branched filaments polymerized by Arp2/3 and generated by WASP[45]. WASP is involved in the differentiation of pTreg cells and in the self-antigen-driven activation of pTreg cells[65]. However, the upstream signaling pathways associated with the formation of actin foci have not been explored as Cdc42 has been reported to be unnecessary for the TCR-dependent localization of WASP in actin foci[66]. In this study, Rap1-GTP is suggested to be necessary for WASP localization at actin foci. RIAM is Rap1-GTP-binding protein and a member of the MRL family of proteins, and a regulator of integrin activation and cytoskeletal organization[67,68]. RIAM is also reported to be involved in TCR-mediated signaling[52]. In addition, RIAM is reported to be associated with SKAP55(Src kinase-associated phosphoprotein of 55 kD)[69]. SKAP55 is associated with SLP76 MCs and critical for integrin-independent adhesion via the TCR[70]. However, RIAM did not localize at actin foci, but accumulated at ruffling membrane in response to TCR engagement. The deletion of RIAM in 3A9 T cells induced morphological changes, but did not inhibit the formation of actin foci after the stimulation with anti-CD3. Rap1-GTP-binding proteins, RAPL (Rap1-associated adhesion and polarization of lymphocytes) and Mst1(mammalian STE20-like kinase 1) were downstream effectors of Rap1, and play critical roles in LFA-1 adhesion[61,63]. However, RAPL- or Mst1-deficient T cells did not show defects in TCR-mediated signaling and IL-2 production by the crosslinking of TCR with anti-CD3. In the present study, we found that Rap1 could associate with WASP in a GTP-independent manner. However, it remains unexplored whether defects in TCR-signaling by Rap1-deficiency is due to only impaired formation of WASP-dependent actin foci, and whether Rap1 is directly or indirectly involved in WASP localization at MCs of SLP-76. The role of Rap1 in the recruitment or activation of WASP is an interesting topic that remains to be elucidated.

In this study, the proportion and numbers of RORγt− (Helios+) Treg cells rather increased after the onset of colitis. RORγt− (Helios+) Treg cells were mainly selected with self and MHC in the thymus, and might be expanded by inflammatory cytokines. Although CTLA-4 expression on both RORγt+ and RORγt− Treg cells were reduced in

Rap1KO mice, our data might support the notes that RORγt+ Treg cells play key suppressor of colitogenic Th17 cells in the LILP.

Previous studies have demonstrated that Rap1 and its downstream effector molecules, RAPL and Mst1, are activated by antigen stimulation through the TCR, which is required for immunological synapse (IS) formation with APCs in an LFA-1-ICAM-1-dependent manner[62,63]. As Rap1 deficiency has been found to impair IS formation, the role of Rap1 as a regulator of actin organization has not been elucidated in the IS. The present study revealed that Rap1 plays indispensable roles in the TCR-dependent formation of actin foci in the IS. Thus, Rap1 plays a central role in the immune responses to antigens and may be a target for treating various immune diseases, such as intestinal bowel disease and autoimmune diseases.

## Methods

**Mice and cells.** All animal experiments were conducted in accordance with the Regulations for the Care and Use of Laboratory Animals at Kitasato University, and the protocols used in the present study were ethically approved by the Institutional Animal Care and Use Committee at Kitasato University. *Rap*1a$^{f/f}$ mice containing floxed exons 2–3 of *Rap1a* and *Rap1b*$^{f/f}$ mice containing floxed exon 1 of *Rap1b* on a C57BL/6J background were maintained under specific pathogen-free conditions. These mice were crossed with CD4-Cre mice, yielding mice with T-cell-specific *Rap1a/b* deletion. IL-17 knockout mice[71] were crossed with T-cell-specific *Rap1a/b* mice. Mice, both females and males, were used for the experiments at 2.5~11 weeks of age. For eradication of intestinal bacterial flora, Rap1KO mice were treated with a cocktail of antibiotics containing streptomycin (5 g/l), ampicillin (1 g/l), vancomycin (0.5 g/l), neomycin (1 g/l) and 2.5% (wt/vol) sucrose (Wako) in the drinking water starting 5 d before birth and continuing until analysis. Antibiotics treatment was renewed every week.

CD4+ T cells and Treg cells were purified from the spleen of WT and Rap1KO mice using the Naïve CD4+ T Cell Isolation Kit or the CD4+CD25+ Regulatory T Cell Isolation Kit (Milteny Biotec), or sorting using MoFlo XDP (Beckman Coulter). CHO K1 cells were suspended in Ham's F-12 medium containing 10% fetal bovine serum (FBS) and penicillin-streptomycin solution. Moreover, 3A9 hen egg lysozyme (HEL)-specific, *I-A$^k$*-restricted T-cell hybridoma were suspended in RPMI 1640 medium containing 10% FBS and penicillin-streptomycin solution.

**Antibodies and reagents.** Purified anti-CD3; purified anti-CD28; fluorescein isothiocyanate (FITC)-conjugated anti-CD3, -CD19, -B220, -F4/80, -CD64, -NK1.1, -CD62L, -Helios, -Foxp3, -IFNγ; phycoerythrin (PE)- conjugated anti-CD8, -CD11c, -CD44, -Foxp3, -RORγt, -GATA3, -IL-17F; phycoerythrin -cyanin 7 (PE-Cy7)-conjugated anti-CD3, -CD11b, -CD44, -CTLA-4, -T-bet, -IL-17A; allophycocyanin-conjugated anti-CD4, -CD44, -CD62L, -MHCII, -Foxp3; Brilliant Violet$^{TM}$ 421 (BV421)-conjugated anti-CD4, -CD45.1, -CD80, -CD86, -IL-10; Brilliant Violet$^{TM}$ 711 (BV711)-conjugated anti-CD3,-CD45.2, -CD103 (Biolegend or e-bioscience or BD), anti-STIM-1, -NFAT1, -ZAP70, −pZAP70, -SLP76, -pSLP76, -PLCγ, -pPLCγ, -c-Jun, -p-c-Jun, -Myc, -ERK, -p-ERK, -HS1, -pHS1; peroxidase-conjugated goat anti-mouse IgG, peroxidase-conjugated goat anti-rabbit IgG (Cell Signaling), Alexa Fluor 633–conjugated anti-rabbit IgG; and Alexa Fluor 633–conjugated anti-mouse IgG (Invitrogen), anti-pPLCγ (Abcam), anti-

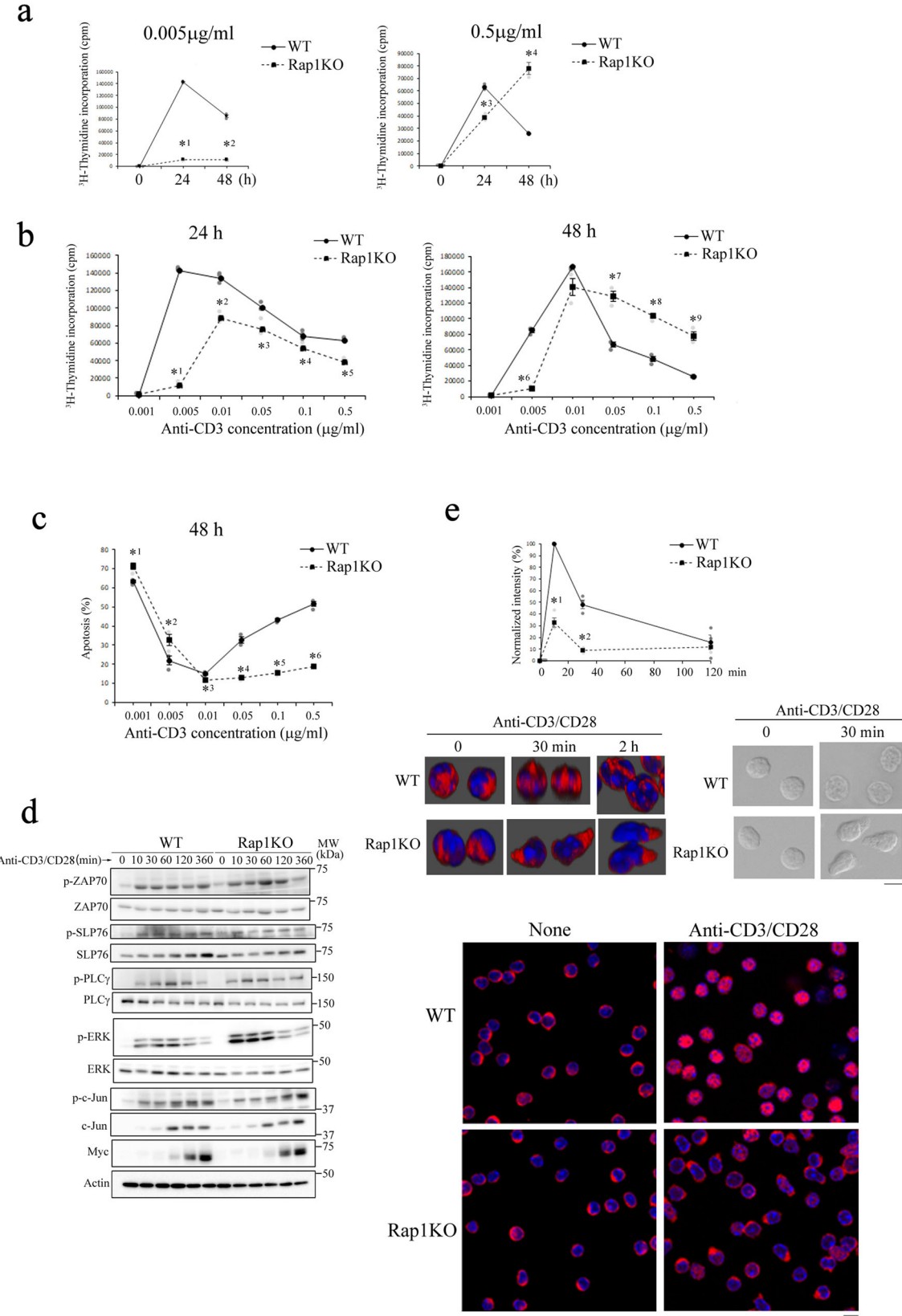

WASP (Santa cruz), Anti-RIAM (Abcam), and Lpd (Invitrogen), were used for flow cytometry, immunostaining and immunoblotting.

**Flow cytometry and cell sorting**. Immunofluorescence flow cytometry was performed as described previously[3]. For mAb staining, the cells were washed with staining buffer (1% FBS in HBSS), resuspended in 50 µl of the same buffer, pre-

incubated with purified anti-mouse CD16/32 (Biolegend) for 10 min, and incubated for 30 min at 4 ˚C with each fluorescence-conjugated mAb or isotype control matched with primary antibody. Zombie NIR™ dye (Biolegend) was used to assess live or dead status of cells. The samples were measured using a Gallios flow cytometry or CytoFLEX (Beckman Coulter). Doublets were distinguished from single cells by plotting FSC height vs FCS area. CD4+ T cells or Treg cells were sorted using a MoFlo XDP (Beckman Coulter). The purity of the sorted

**Fig. 6 Rap1 played crucial roles in TCR-mediated signaling. a** [3H]-Thymidine uptake by WT and Rap1-deficient naïve CD4+ cells. WT and Rap1-deficient naïve CD4+ cells were stimulated with 0.005 µg/ml (left) or 0.5 µg/ml (right) of anti-CD3 in the presence of anti-CD28 for 0, 24, and 48 h. [3H]-Thymidine uptake was measured in triplicate. Data represent the mean ± S.E.M. *1$P < 0.001$ and *2$P < 0.001$, *4$P < 0.001$ and *5$P < 0.001$ compared with WT naïve CD4+ cells. **b** WT and Rap1-deficient naïve CD4+ cells were stimulated with the indicated concentrations of anti-CD3 in the presence of anti-CD28 for 24 (left) and 48 h (right). [3H]-Thymidine uptake was measured in triplicate. Data represent the mean ± S.E.M. *1$P < 0.001$, *2$P < 0.001$, *3$P < 0.02$, *4$P < 0.03$, *5$P < 0.001$, *6$P < 0.001$, *7$P < 0.002$, *8$P < 0.001$, and *9$P < 0.001$ compared with WT naïve CD4+ cells. **c** Effects of Rap1 deficiency on apoptosis. WT and Rap1-deficient naïve CD4+ cells were stimulated with the indicated concentrations of anti-CD3 in the presence of anti-CD28 for 48 h. Apoptosis was measured by the incorporation of propidium iodide ($n = 3$). Data represent the mean ± S.E.M. *1$P < 0.02$, *2$P < 0.05$, *3$P < 0.008$, *4$P < 0.001$, *5$P < 0.001$, and *6$P < 0.001$ compared with WT naïve CD4+ cells. **d** Effects of Rap1 deficiency on early signaling. WT and Rap1-deficient naïve CD4+ cells were stimulated with 0.5 µg/ml of anti-CD3 in the presence of anti-CD28 for the indicated times. Total lysates from the stimulated cells were immunoblotted for anti-phosphorylated ZAP-70, -SLP76, -PLCγ, -ERK, and -c-Jun, and anti- ZAP-70, -SLP76, -PLCγ, -ERK, -c-Jun, -Myc and actin. **e** (Upper) Defective Ca2+ influx in Rap1-deficient naïve CD4+ cells. WT and Rap1-deficient naïve CD4+ cells were loaded with Fluo4, incubated on a plate coated with anti-CD3 and anti-CD28, and monitored for Ca2+ influx at the indicated times, as described in the Methods. The Ca2+ influx in WT and Rap1-deficient naïve CD4+ cells is presented as the fold change in the normalized MFI of Fluo4 in WT or Rap1- deficient naïve CD4+ cells stimulated for the indicated times relative to the peak MFI in WT naïve CD4+ cells (adjust 100) ($n = 3$). Data represent the mean ± S.E.M. *1$P < 0.001$ and *2$P < 0.002$ compared with WT naïve CD4+ cells. (Middle) Three-dimensional localization of STIM-1 (left) and the morphology (right) of WT and Rap1-deficient naïve CD4+ cells stimulated with anti-CD3 and anti-CD28 for 30 min or 2 hr. Scale bar, 5 µm. (Lower) Nuclear translocation of NFAT1 in WT and Rap1-deficient naïve CD4+ cells. WT and Rap1-deficient naïve CD4+ cells were incubated on glass coated with anti-CD3 and anti-CD28 for 6 h at 37 °C, fixed, and immunostained for NFAT1 (red) and DAPI (blue).

populations was more than 95%, as determined by a presorted sample run in parallel. Data were analyzed using in Kaluza analysis version 2.1 (Beckman Coulter).

**Immunoblot analysis**. T cells were stimulated with immobilized anti-CD3 and anti-CD28 for 0, 10, 30, 120, 360 min, then lysed in buffer (1% Nonidet P-40, 150 mM NaCl, 25 mM Tris-HCl [pH 7.4], 10% glycerol, 2 mM MgCl$_2$, 1 mM phenylmethylsulfonylfluoride, 1 mM leupeptin, and 0.1 mM aprotinin). Cell lysates were subjected to immunoblotting.

**Histological examination**. LILP sections from WT, IL-17AKO, Rap1KO, DKO mice were fixed in 10% buffered formalin and embedded in paraffin. Paraffin-embedded LILP sections were cut (6-µm), stained with hematoxylin and eosin (H&E) and examined using an Olympus IX51 light microscope equipped with CCD camera. Histological grades were assigned in a blinded manner as previously described[4].

**Induced culture of bone marrow-derived dendritic cells**. Bone marrow cells were isolated and cultured in RPMI-1640 medium supplemented with 10% FBS at a density of $1 \times 10^6$ cells/ml. Subsequently, GM-CSF was added to the medium to a final concentration of 10 ng/ml. The culture medium was replaced 48 h later to remove unattached cells and cell debris. Following this, fresh medium was supplemented with GM-CSF. On day 7, semi-suspended cells and loosely attached cells were collected by gently pipetting the medium against the plate. The cells were further incubated with lipopolysaccharide (LPS) for 24 h, and BMDCs were obtained. More than 90% of the cells were confirmed to be CD11c+ and expressed CD86 and CD80.

**In vitro differentiation of CD4+ cells**. Naïve CD4+ cells were isolated and cultured on anti-CD3- and anti-CD28-coated plates with cytokines and blocking antibodies; the details are as follows: Th1 cells: 5 ng/ml rmIL-12 and 5 ng/ml hIL-2; Th2 cells: 10 ng/ml rmIL-4, 5 ng/ml hIL-2, and 1 µg/ml soluble XMG1.2; iTreg cells: 1 ng/ml hTGFβ and 1 µg/ml XMG1.2; and Th17 cells: 20 ng/ml rmIL-6, 2 ng/ml hTGFβ, and 1 µg/ml XMG1.2. The cells were incubated at 37 °C with 5% CO$_2$ for 3 to 4 days before analyzing transcription factor expression and cytokine production. For cytokine analysis by flow cytometry, the cells were restimulated with phorbol myristate acetate (PMA; 50 ng/ml) plus ionomycin (500 ng/ml) for 4–6 h in the presence of a Golgi inhibitor, such as brefeldin A, and intracellular cytokine staining was performed.

**Intracellular cytokine staining**. For the analysis of intracellular IFNγ, IL-10, IL-17A, and IL-17F, cells were stimulated for 4 h in IMDM containing 50 ng/ml PMA, 500 ng/ml ionomycin, and 10 µg/ml brefeldin A. After surface staining, the cells were fixed, permeabilized, and intracellularly stained with the FOXP3/Transcription Factor Fixation/Permeabilization Concentrate and Diluent solution (e-bioscience) and antibody, according to the manufacturer's protocol. Data were collected by flow cytometry.

**In vitro Treg cell suppression assay**. Responder WT naive CD4+ cells were labeled with 1 µM 5, 6-carboxyfluorescein diacetate (CFSE, DOJINDO). The labeled responder cells ($5 \times 10^5$ cells/ml) were cultured with 1µg/ml of anti-CD3 in the presence of BMDC with or without WT or Rap1-deficient Treg cells. On day3, the cells were stained with anti-CD4 and -CD25, then analyzed by flow cytometry[72].

**[3H]-Thymidine incorporation assay**. Purified T cells were plated into 96-well plates in triplicates and stimulated with 0.001, 0.005, 0.01, 0.05, 0.1, 0.5 µg/ml of anti-CD3 and 2.5µg/ml of anti-CD28 for 24–48 h. In total, 1 mCi [3H]-thymidine was added 6 h before harvest. Labeled DNA from the cells was collected on GSC filters (Whatman), and the radioactivity was measured in a scintillation counter.

**Isolation of cells from LILP**. The intestines were opened longitudinally, washed in PBS to remove the intestinal content, and then cut into 1 cm pieces. The pieces were placed in PBS with 30 mM EDTA, incubated at 4 °C for 15 min, and then washed with PBS by vigorous shaking for four cycles. The washed intestinal pieces were cut into 1 mm pieces and incubated for 1 h at 37 °C in DMEM containing 1 mg/ml collagenase D and 0.2 mg/ml DNase I. Every 15 min, tissues were pipetted and reincubated in fresh medium containing collagenase D and DNase I; supernatants were collected at each step. The supernatants were centrifuged and resuspended in 40% Percoll (GE Healthcare). Mononuclear cells were collected from interphase of 80 and 40% Percoll after centrifugation at 2800 rpm for 15 min.

**In vitro TE assay**. CD80-GFP CHO cells were generated by transduction with lentiviral vectors (CSII-EF-MCS) carrying CMV promotor-driven eGFP-fused mouse CD80[29]. The resulting transfectants were sorted (MoFlo XDP, Beckman Coulter) for uniform GFP expression. Naïve CD4+ T cells isolated from the spleen of WT and Rap1KO mice were mixed at a 1:1 ratio with CD80-GFP donor CHO cells for 24 h in the presence of soluble anti-CD3 (5 µg/ml) or PMA (12.5 ng/ml)/ionomycin (100 ng/ml) and TAPI2 (100 µM, Cayman Chemical); BafA (25 nM, AdipoGen) was used for up to 6 h before culture termination. The mixed cells were stained with anti-CD4 and anti-CTLA-4 and analyzed by flow cytometry. The mixed cells were immobilized on poly-L-lysine-coated slides, and confocal images (TCS SP8, Leica) were obtained using a 63x objective lens.

**Confocal microscopy and time-lapse imaging**. Immunostaining and live-cell imaging were performed as previously described[73]. CD4+ cells were incubated with immobilized anti-CD3 and anti-CD28 for 0–30 min in Culture Slides (Corning). After incubation, the cells were fixed and stained with phalloidin, and anti-STIM-1, -NFAT-1, -WASP, -pHS-1, -pZap70, and/or -pSLP76. Confocal images (TCS SP8, Leica) were obtained using a 63x objective lens. Time-lapse confocal images were also obtained in the multitrack mode. Line profiles of the confocal images were obtained using the software ImagePro (MediaCybernetics).

**Ca2+ measurement**. The fluorescent dye Fuluo4-AM (DOJINDO), a Ca2+ indicator, was used to quantify [Ca2+]i in T cells by flow cytometry. WT and Rap1-deficient naïve CD4+ cells were loaded with 5 µM Fuluo4-AM in IMDM (Wako) containing 5% FBS and 0.04% Pluronic F-127 (Sigma) for 30 min at a concentration of $1 \times 10^6$ cells/ml and then washed twice with fresh medium. The cells were incubated with immobilized 0.5 µg/ml anti-CD3 and 2.5 µg/ml anti-CD28 for 0, 10, 30, 120 min. [Ca2+]i concentrations in T cells were measured by flow cytometry.

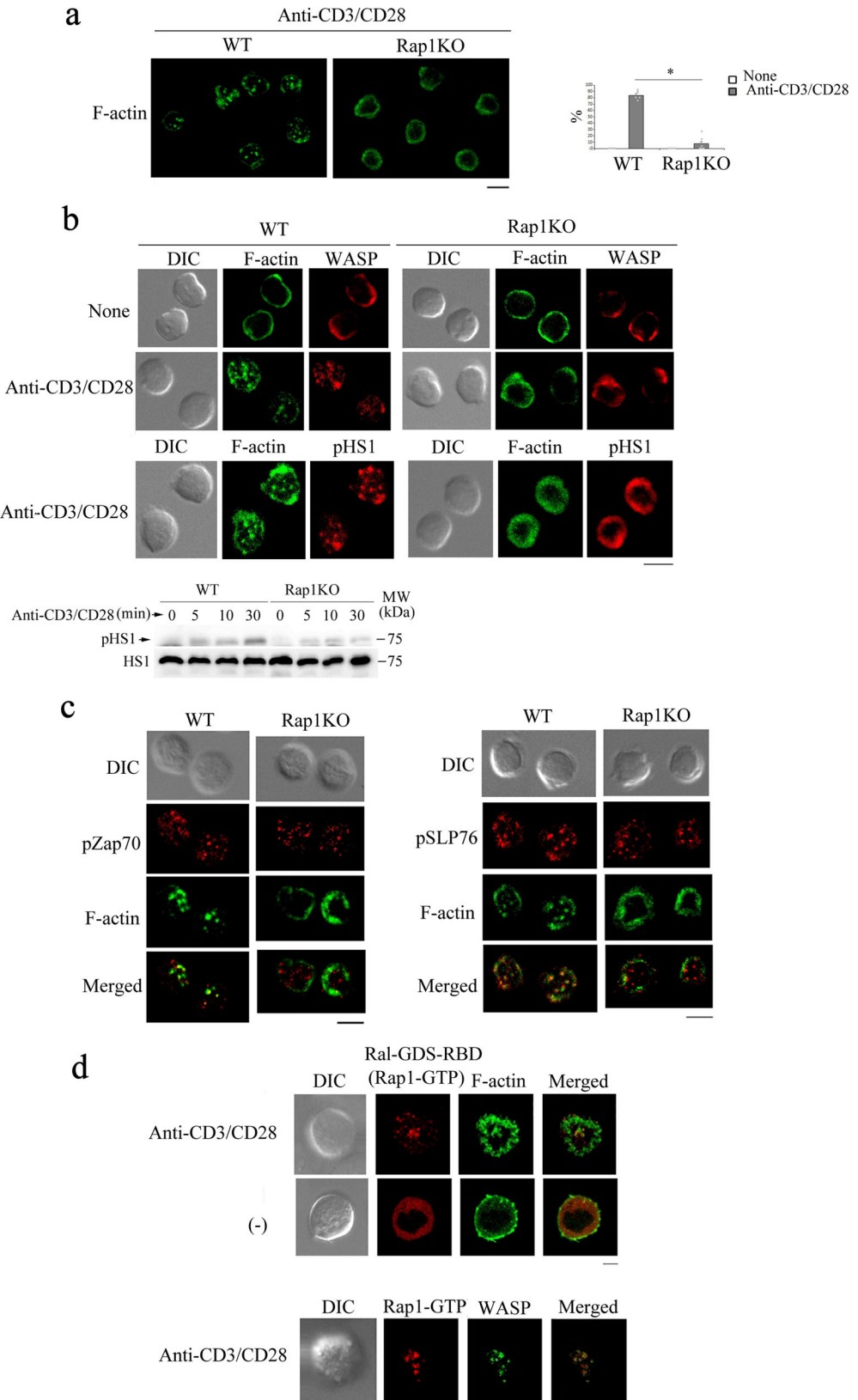

**Pull-down assay**. Halo-tagged WASP was pulled down with GST fused to the Rap1A which deleted C-terminal[74]. In brief, COS cells were transfected with the plasmid carrying Halo-tagged WASP, lysed in ice-cold lysis buffer (1% TritonX-100, 50 mM Tris-HCl [pH 7.5], 100 mM NaCl, 10 mM MgCl$_2$, 1 mM phenylmethylsulfonyl fluoride, 1 mM leupeptin, and 0.5 mM aprotinin), and pulled down with GDP-loaded or GTPγS-loaded Rap1-GST coupled to glutathione agarose beads for 1 h at 4 °C. The beads were washed twice with lysis buffer and then subjected to immunoblotting.

**Fig. 7 Rap1 was involved in the formation of actin foci. a** (Left) Defective actin polymerization in Rap1-deficient naïve CD4[+] cells. WT and Rap1-deficient naïve CD4[+] cells were incubated on glass coated with anti-CD3 and anti-CD28 for 10 min, fixed, stained for F-actin using Alexa488-phalloidin (green), and imaged using confocal microscopy. Representative images of three experiments are shown. Scale bar, 5 μm. (Right) Graphs represent the mean ± S.E.M. of the percentages of WT and Rap1-deficient naïve CD4[+] cells showing F-actin clusters ($n = 30$). *$P < 0.001$ compared with WT naïve CD4[+] cells. **b** (Upper) Defective F-actin foci in Rap1-deficient naïve CD4[+] cells. WT and Rap1-deficient naïve CD4[+] cells stimulated with anti-CD3 and anti-CD28 for 10 min were stained with Alexa488-phalloidin (green) and anti-WASP or anti-phosphorylated HS1 (red). Representative images are shown. Scale bar, 5 μm. (Lower) Phosphorylation of HS1. WT and Rap1-deficient naïve CD4[+] cells stimulated with anti-CD3 and anti-CD28 for the indicated times were lysed and immunoblotted for anti-phosphorylated HS1 and total HS1. **c** Normal formation and localization of MCs, including phosphorylated ZAP70 and phosphorylated SLP76, in Rap1-deficient naïve CD4[+] cells. WT and Rap1-deficient naïve CD4[+] cells stimulated with anti-CD3 and anti-CD28 for 10 min were stained with Alexa488-phalloidin (green) and anti-phosphorylated ZAP70 (left) and SLP76 (right) (red). Representative images are shown. Scale bar, 5 μm. **d** Localization of Rap1-GTP in actin foci. 3A9 T cells expressing Ral-GDS-RBD-mCherry, a reporter of Rap1-GTP, and Lifeact-GFP, a reporter of F-actin (upper) or WASP-GFP (lower), were stimulated on glass coated without or with anti-CD3 and anti-CD28 at 37 °C for 5 min, then the digital images of contact areas of the cells were taken. Scale bar, 5 μm.

**Adoptive cell transfer**. Rap1KO mice (Ly5.2[+]) of 6 weeks old were injected intravenously with $10^6$ naïve CD4[+]CD25[-]CD62L[+]CD44[−] T cells which were derived from LNs of 7-8 weeks old WT congenic mice (Ly5.1[+]). The recipient mice were observed daily and weighed every 2 or 3 days, and sacrificed 4 weeks later to examine T cell population in the LILP and the severity of colitis by histological study.

**DNA constructs and transfection**. To produce a mammalian expression vector of mCherry-tagged Ral-GDS-RBD, we subcloned cDNAs encoding mCherry and the RBD of human Ral-GDS (amino acids 772–868) into a lentivirus vector (CSII-EF-MCS; a gift from H. Miyoshi, RIKEN, Wako, Japan). *WASP* cDNA transferred to a pFN21A vector was purchased from Kazusa, and we subcloned cDNA encoding WASP and GFP into a lentivirus vector (CSII-EF-MCS). Lifeact-GFP was constructed as previously described[73], and subcloned into a lentivirus vector (CSII-EF-MCS). The fidelity of all the constructs was verified by sequencing. The constructs were transduced to 3A9 T cells by lentivirus.

**RNA-mediated interference and gene introduction via lentiviral transduction**. RNA-mediated interference was used to suppress mouse Rap1a and Rap1b expression. A 19-nucleotide -specific sense RNA sequence of *Rap1a* (GAAT GGCCAAGGGTTTGCA) (5'–3') and *Rap1b* (AGACACTGATGATGTTCCA) (5'–3') or a scrambled control RNA sequence were introduced into BAF/LFA-1 using lentivirus with a lentivirus vector with or without GFP or puromycin resistance gene (a gift from Dr. Miyoshi H., RIKEN, Wako, Japan) containing the RNAi construct under control of the H1 promoter cassette, respectively. The transduction efficiencies were greater than 90%. A GFP high population was collected by cell sorting and subjected to adhesion assays and immunoblot analysis.

**Generation of RIAM-deficient 3A9 T cells by the CRISPR/Cas9 system**. The guide sequence targeting exon of the mouse RIAM was cloned into pX330 (Addgene #42230)[75]. pX330-U6-Chimeric_BB-CBh-hSpCas9 was a gift from Feng Zhang (Addgene plasmid # 42230; RRID: Addgene_42230). pCAG-EGxxFP was used to examine efficiency of the target DNA cleavage by the guide sequence and Cas9 activity. The resultant guide sequence of exon 3 (GTGTAGT-TAAACTCTTCTCG) (5'–3') was cloned into GFP expressing plasmid DNA pX458 (Addgene #48138)[76]. pSpCas9(BB)-2A-GFP (PX458) was a gift from Feng Zhang (Addgene plasmid # 48138; RRID: Addgene_48138). The pX458 plasmid was transfected into 3A9 T cells. 24 h after transfection, cells were sorted GFP-high population, followed by limiting dilution. Expression of full length RIAM protein in each isolated clone was tested by immuonblotting. Exon of RIAM from edited clones was PCR amplified and verified by sequencing.

**16S rRNA gene sequencing and analysis of murine cecal samples**. Cecal samples were immediately frozen after collection. They were kept in a freezer at −80 °C for further analyses. The 16S rRNA gene was analyzed as previously described with some modifications[77]. First, cecal samples were lyophilized for ~12–18 h using a VD-800R lyophilizer (TAITEC, Nagoya, Aichi, Japan). Each freeze-dried cecal samples were crushed and weighed ~10 mg. Second, the sample was combined with four 3.0-mm zirconia beads, ~100 mg of 0.1-mm zirconia/silica beads, 300 μL DNA extraction buffer (TE containing 1% (w/v) sodium dodecyl sulfate), and 300 μL of phenol/chloroform/isoamyl alcohol (25:24:1). Then, mixed samples were subjected to vigorous shaking (1500 rpm for 15 min) using a Shake Master (Biomedical Science, Shinjuku, Tokyo, Japan). The resulting emulsion was subjected to centrifugation at 17,800×g for 10 min at room temperature. RNA was removed from the sample by RNase A treatment from bacterial genomic DNA purified from the aqueous phase. The resulting DNA sample was then purified again by another round of phenol/chloroform/isoamyl alcohol treatment and ethanol precipitation by GENE PREP STAR PI-480 (Kurabo Industries Ltd., Osaka, Japan). V1–V2 hypervariable region of 16S rRNA encoding genes were amplified by PCR using bacterial universal primer set[78,79]. The amplicons were analyzed using a MiSeq sequencer (Illumina, San Diego, California, U.S.A.) with some modifications previously indicated[77]. Filter-passed reads were processed using the Quantitative Insights into Microbial Ecology (QIIME) 2 (2019.7)[80]. Denoising and trimming of sequences were processed using DADA2. 23 bp and 22 bp reads were trimmed from 5′ ends of forward and revers reads respectively to remove primer sequence. 283 bp and 213 bp length reads from 5' ends were used for further steps. Taxonomy was assigned using the SILVA132[81,82] database using the Naive Bayesian Classifier algorithm. The microbiome analysis data have been deposited at the DDBJ Sequence Read Archive (http://trace.ddbj.nig.ac.jp/dra/) under accession number DRA012941.

**Statistics and reproducibility**. All assays were carried out in triplicate or more. Statistical analysis was performed using analysis of variance (ANOVA) followed by the Turkey-Kramer comparison test (Figs. 1c, d and 2b and Supplementary Figs. 1c and 5c) and two-tailed Student's *t* test (Figs. 1a, b, 2a, b, c, 3a, b, c, 4–6 and Supplementary Figs. 1b, 2–4). *P* values less than 0.05 were considered significant.

**Reporting summary**. Further information on research design is available in the Nature Research Reporting Summary linked to this article.

## Data availability
Data generated or analyzed during this study are included in this published article and Supplementary Data 1. The sequence data are deposited at the DDBJ Sequence Read Archive (http://trace.ddbj.nig.ac.jp/dra/) under accession number DRA012941. Any remaining information can be obtained from the corresponding author upon reasonable request.

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

## Acknowledgements

We would like to thank Ms. M. Mamiyoda and Ms. R. Sugioka for technical assistance. This work was supported by Japan Society for the Promotion of Science KAKENHI 20K16290 (to K.K.) and 19K07612 (to S.I.), Takeda Science Foundation (to S.I.), The Naito Foundation (to S.I.), Kitasato University Research grant for young researchers (to S.I.), Lilly Foundation (to K.K.), Morinaga Foundation (to S.I.), JST ERATO (JPMJER1902 to S.F.), AMED-CREST (JP20gm1010009 to S.F.), and the Takeda Science Foundation (to S.F.) and the Food Science Institute Foundation (to S.F.)

## Author contributions

S.I. and K.K designed, performed experiments, and wrote the paper. T.S. H.M. and H.Y. performed the experiments. N.F., S.F. and T.Y. contributed to the preparation of essential materials and commented on the experiments and paper.

## Competing interests

The authors declare no competing interests.
