## [Peer Review File · Communications Biology]

Reviewers' comments:

Reviewer #1 (Remarks to the Author):

In the manuscript titled "Role of Rap1 in preventing colitogenic Th17 cell expansion and in Treg cell differentiation and distal TCR signaling" Ishihara et al., report that Rap1 deficiency in T cells impairs development or maintenance of Rorg+ Tregs in the colonic lamina propria. This reduction in Rorg+ pTregs is accompanied by subsequent increase in pathogenic TH17 cells and leads to development of spontaneous colitis. The authors propose impaired TCR engagement coupled with reduced CTLA4 expression as a possible cause of reduced Treg function in RAP1KO mice. The study is novel and identifies a possible Treg dependent role for Rap1 in intestinal inflammation and colitis. There are some points that should be addressed prior to publication:

1. In figure 1, the authors report IL17 levels as an inflammatory read out. Increased IFN γ production is also present in human and mouse colitis and Rorg+ Tregs have been shown to control TH1 cells or IFN γ producing pathogenic TH17 cells? Do Rap1KO mice have high expression of IFN γ and is this reduced by antibiotics?
2. Rorg+ Treg proportions are reported out of total CD4+ T cells. Given a substantial increase in TH17 cells which may affect overall frequencies of Tregs, authors should report Rorg+ Treg numbers and frequencies out of Foxp3+ cells.
3. The authors show antibiotics resolve intestinal inflammation but do not present any data in microbial composition of Rap1KO, WT, DKO or antibiotic treated mice. Is spontaneous colitis due to enrichment of certain microbes, pathobionts or just a decrease in overall microbial load etc? It would be helpful to characterize the microbiota using 16S seq in various condition pre and post-antibiotics.
4. CTLA-4 deficiency or blockade has been reported to have a similar role outcome as in Rap1. In these studies, CTLA-4 on Tregs but not T conv cells is the cause of inflammation (Lui et al., JI 2018). Since CTLA4 expression is comparable in Rorg+ vs Rorg- Tregs and a similar reduction is seen in both in Rap1KO, the specific role of CTLA-4 in Rorg+ Tregs in Rap1KO is not substantiated. Maybe adoptive T cell transfer assays of sorted CTLA4 deficient vs sufficient Rorg+ vs Rorg- Tregs (and Rap1 deficient vs sufficient Rorg+ vs Rorg- Tregs) in combination with various deficient conventional cells or Rorg blockade with small inhibitors could be helpful.

The manuscript has appropriate statistical analysis and power.

Reviewer #2 (Remarks to the Author):

In their manuscript "Role of Rap1 in preventing colitogenic Th17 cell expansion and in Treg cell differentiation and distal TCR signaling" Ishihara et al evaluated the role of Rap1 ko specifically in T cells for colitogenic Th17 cell expansion, intestinal Treg differentiation and distal TCR signaling. The authors have previously described the ko mice and found spontaneous development of colitis due to altered Th1/Th17 cell migration into the gut. In the present study they describe that Rap1 deletion in T cells causes exacerbated intestinal Th17 cell expansion which may be due to reduced capacity of Treg differentiation in the intestine. They also show altered TCR signaling which strongly affects in vitro T cell proliferation, differentiation and survival. Overall I have the impression that both points seem to be interesting but have not been carefully evaluated. Especially it is not clear to me how these two points are connected. Therefore I think that the manuscript is not suitable for publication in its present stage and would require significant additional experiments.

1. Fig 1 Authors show increased IL-17 and IL-17/IFN γ producing T cells in the colon of Rap1 ko mice, which is completely prevented by antibiotics treatment. Ko mice develop colitis which is ameliorated by antibiotics treatment. Ko of IL-17A also ameliorates colitis but Th17 development (ROR γ T+ T cells) was maintained.

Please provide example dot plots of RORgT staining within Teffector and Tregs as well as quantification of cell numbers in LP T cells. Spontaneous colitis development and Th17 accumulation has been described by the group previously. The figure is not really essential and can be combined with results from figure 3 addressing the role of Treg differentiation.

2. Fig.2 In vitro Th17 differentiation is not altered regarding RORgT expression but Rap1 ko cells have less IL-17.

Under Treg inducing conditions the authors claim reduced Foxp3 induction with low CD3 stimulation but increased Foxp3 with strong CD3 signal due to a lack of apoptosis in ko versus WT cells.

These Treg data are not convincing and largely unclear: the gates for Foxp3 expression cut right through the population, the effect of the culture conditions on expansion/apoptosis are unclear since no data are provided, e.g. absolute cell numbers, staining for apoptosis. It remains unclear to me what the in vitro conditions should indicate and what exactly happens to the cells.

Also data in Th1 and Th2 cells are incomplete (cell numbers, expansion, cytokine production, ...). It remains unclear how the ko affects activation, proliferation, differentiation. Especially in light of the results from figures 5&6 addressing TCR signalling.

3. Figure 3. Rap1 ko show decreased frequencies of LP RORgT Foxp3 cells, whereas RORgT Foxp3- are increased. This is not affected by IL-17A ko, i.e. not induced by inflammation. Transferred WT T cells preferentially develop into RORgT Treg and transfer prevents colitis. These data are potentially interesting pointing to an Treg intrinsic role of Rap1 in the intestine. All data should be condensed into 1 main figure rather than into suppl material. The data suggest that Rap1 ko Treg cannot prevent colitis. However, the transfer system CD25 T cells does not allow to conclude that Tregs were induced from naïve non-Treg precursors. It would be important to show that by generating a Foxp3cre specific Rap1 ko to demonstrate an Treg intrinsic effect. Alternatively a colitis transfer model using mixed transfers of ko and WT cells can be used. I also find the gating in figure 4c not ideal: gate on all transferred or resident CD4 T cells – display Foxp3 versus RORgT (to visualize the Treg/Tcon ratio) second gate on Foxp3 pos or neg: display GATA3 (or Helios) versus RORgT and another dot plot for cytokines by both subsets.

4. In figure 4 the authors show that Treg from Rap1 ko have reduced capacity to suppress to reduced CTLA4 expression and transendocytosis in response to TCR but not PMA/iono stimulation. It is not clear to me how this has been done, gating and calculations. 4 b) & d) suppression assay and transendocytosis assay: how have the Tregs been sorted, from which source and is the purity comparable between ko and WT? 4c) staining and gating of DC subsets?

5. In figure 5 a TCR signaling defect is demonstrated which affects T cell activation, proliferation and apoptosis in response to titrated amounts TCR stimulus. These data should be combined with data from figure 2. The overall message of these results for the interpretation of the ko phenotype are not clear to me.

6. In figure 6 the authors suggest that Rap1 ko T cells have a defect in TCR actin formation. This is not my expertise and I cannot judge the quality of the data.

However, I have the impression that for data in figure 5&6 it is not clear how this affects Tregs and how these data are connected to the in vitro and vivo Treg phenotype and the development of colitis.

Reviewer #3 (Remarks to the Author):

The manuscript by Katagiri and co-workers describes a novel role of the Rap-1 GTPase in the differentiation of intestinal RORgt+ Treg cells. They use T cell-specific Rap1 ko mice as a model system and find that the Th17 T cell population is strongly increased in their large intestinal lamina propria. This was microbiota-dependent, and IL-17A was found to be responsible for the induction of colitis in these animals. Rap1 does not play a direct functional role in Th17 cells, however, but the absence of Rap1 expression much rather impedes the differentiation of RORgt+ Treg cells to a

high degree. To approach a mechanistic workup of these findings, the authors find that CTLA-4 expression on Rap1-deficient Treg cells is lower than on their WT counterparts, and thus CD80 and 86 expression, respectively, was found elevated. Employing in vitro studies, they could further demonstrate that Rap-1 deficient naïve CD4 cells showed impaired CTLA-4 induction, defective nuclear translocation of NFAT, and defective formation of actin foci in response to TCR engagement.

The study is a confirmation and strong extension of previous findings regarding the role of Rap-1 in Treg differentiation. The description of a selective and important functional role of Rap-1 in gut-specific Treg is entirely novel, and of course broadly interesting. The paper is also in general well written and presented. However, exactly how Rap-1 is mechanistically involved in driving T cell differentiation polarity is not so clear.

I have two main issues with this. One is regarding the role of the integrin adhesion receptor LFA-1. Rap-1 and its substrates are important mediators of LFA-1 conformational activation and the authors are eminent experts of deciphering molecular regulation of adhesion receptors in T cells. However, they merely cite older work of others to claim that the new Rap-1 functions described here are LFA-1-independent. I think this needs further clarification and should at least be discussed more precisely and to the point. LFA-1 is such an important player in immune synapse formation – this needs more illumination, particularly by these authors.

The other concern is regarding the mechanistic analyses part of the paper. They are novel and interesting, but remain somewhat descriptive and do not really explain a lot. Where does the specificity of Rap-1 function in Treg regulation and more selectively in RORgt+ Treg differentiation come from? According to the authors, this is due to downregulation of CTLA-4 expression and the resultant lack of CD80/86 transendocytosis? How could this work? Is CTLA-4 selectively associated or co-localized with Rap-1 or its substrates? How does the WASP association of Rap-1 aid in this? I can certainly see that the whole workup of this issue would result in substantial more experimentation, but the presented explanations and associations are nonetheless rather loose and vague. Also, and importantly: if CTLA-4 really is the direct and specific target of Rap-1 function in this context, then the regulatory impact of Rap-1 should not be restricted to Tregs, because CTLA-4 is not a Treg-specific molecule. How about CTLA-4 expression and CD80/86 transendocytosis on Rap-1^{-/-} effector cells, including IL17 expressing cells? And with respect to Tregs: does Rap1 deficiency affect Foxp3 expression (via NFAT inactivation)? What is the relation of actin foci regulation with respect to CTLA-4 regulation and CD80/86 transendocytosis?

Minor point:

The classification “Rap1 is involved in distal TCR signaling” seems rather arbitrary. If this relates to a temporal axis, then Rap1-deficiency does not seem to alter phosphorylation responses (Fig 5d) over extended time courses (hours), but it does affect calcium signaling on a rather short time scale (peaks roughly at 10 min after stimulation). The term should thus be defined more precisely or changed.

Reviewer #4 (Remarks to the Author):

In this study, Ishihara et al show the function of small GTPase Rap1 in Th17 and Treg cells. Using a T cell specific Rap1 knockout model, the authors show diminished differentiation of RORgt+ Treg cells in the intestine with a concomitant increase in IL17A, which led to intestinal pathology. Loss of Rap1 also affected CtlA-4 expression in Treg cells and perturbed signalling through the T cell receptor. While the data presented is clear and supports the conclusion, this reviewer wonders why loss of Rap1 specifically affects Treg cells. If this molecule was downstream of TCR signalling in all T cells (CD4/CD8), these mice would also have defect in Th17 cells, which seems not to be the case in vivo.

1. Quantification (graph) for Fig 1b.
2. Fig1c. Spellcheck “antibiotics”
3. In the results section, the authors should mention that Rap1KO is a T cell specific deletion of

Rap1a/b using CD4-cre. Without reading the methods section, this aspect is not obvious.

4. In Fig 2b, gating for Foxp3+ cells should be relaxed. The horizontal line in the quadrant could be lowered to include all Foxp3 expressing cells.

5. In low anti-CD3 Treg polarizing condition, if the gating is adjusted, %Foxp3+ cells might become similar in WT and Rap1Kos. However, there appears to be a clear defect in Foxp3 MFI in the KOs.

6. While CD4+ T cells in the intestine, blood and mesLNs show more IL-17A expression in Rap1KOs, this phenotype seems to be reversed in vitro. The authors should discuss this. Is it simply the lack of Tregs?

7. Is Rap1 a Treg specific molecule? Why does its loss affect Rorgt induction in Tregs but not in CD4 T cells? Or is it a defect in Foxp3 induction in vivo?

8. To understand the differential role of Rap1 in Treg cells and Foxp3-CD4 T cells, the authors should compare the proliferation of Rap1 deficient Tregs and CD4 T cells to WT counterparts.

9. Fig 5d. Western blot needs loading control.

Reviewer #1:

1. In figure 1, the authors report IL17 levels as an inflammatory read out.

Increased IFN γ production is also present in human and mouse colitis and Rorg⁺ Tregs have been shown to control TH1 cells or IFN γ producing pathogenic TH17 cells? Do Rap1KO mice have high expression of IFN γ and is this reduced by antibiotics?

(R) In this study, the proportion and cell numbers of IFN γ (and IL-17)-expressing cells were increased in Rap1KO as shown in Fig.1a and Supplementary Fig.1a. Antibiotics inhibited the production of IFN γ in Supplementary Fig.1a, suggesting that IFN γ production might be involved in pathogenesis.

2. Rorg⁺ Treg proportions are reported out of total CD4⁺ T cells. Given a substantial increase in TH17 cells which may affect overall frequencies of Tregs, authors should report Rorg⁺ Treg numbers and frequencies out of Foxp3⁺ cells.

(R) We added the numbers and frequencies of ROR γ ⁺ Treg out of Foxp3⁺ cells in Fig.2a and Supplementary Fig.2a.

3. The authors show antibiotics resolve intestinal inflammation but do not present any data in microbial composition of Rap1KO, WT, DKO or antibiotic treated mice. Is spontaneous colitis due to enrichment of certain microbes, pathobionts or just a decrease in overall microbial load etc? It would be helpful to characterize the microbiota using 16S seq in various condition pre and post-antibiotics.

(R) We have characterized the microbiota using 16S seq and added those data in Supplementary Fig.1c. We are now going to clarify which microbe causes the colitis using germ-free Rap1KO mice, and will report it as next paper.

4. CTLA-4 deficiency or blockade has been reported to have a similar role outcome as in Rap1. In these studies, CTLA-4 on Tregs but not T conv cells is the cause of inflammation (Lui et al., JI 2018). Since CTLA4 expression is comparable in Rorg⁺ vs Rorg⁻ Tregs and a similar reduction is seen in both in Rap1KO, the specific role of CTLA-4 in Rorg⁺ Tregs in Rap1KO is not substantiated. Maybe adoptive T cell transfer assays of sorted CTLA4 deficient vs sufficient Rorg⁺ vs Rorg⁻ Tregs (and Rap1 deficient vs sufficient Rorg⁺ vs Rorg⁻ Tregs) in combination with various deficient conventional cells or Rorg blockade with small inhibitors could be helpful.

(R) ROR γ ⁺ Treg are predominantly generated from naïve CD4⁺ cells in LILP by 4 weeks of age in a TCR-dependent manner in response to intestinal microbiota, and suppress microbe-specific Th17 differentiation (Lathrop, 2011; Russler-Germain, 2017; Sefik, 2015; Yang 2016; Xu, 2018; Neumann, 2019). ROR γ ⁻ Treg cells are differentiated in the thymus. ROR γ ⁺ Treg cell deficiency is reported to exacerbate colitogenic Th17 cell differentiation and promotes colonic inflammation. In this study, adoptively transferred WT naïve CD4⁺ T cells differentiated into ROR γ ⁺ Treg cells, but not ROR γ ⁻ Treg in the LILP, and prevented the development of colitis in Rap1KO mice (Fig. 3), suggesting the defective functions of ROR γ ⁺ Treg cells might cause the colitis in Rap1KO mice.

We are staining intracellular ROR γ t, and not able to isolate ROR γ ⁻ or ROR γ ⁺ Treg in live condition.

Rap1-deficiency induces severe lymphopenia in lymphoid tissues (Ishihara, 2015), and it is reported that the inhibition of CD4⁺ T cell expansion by Treg in a lymphopenic environment requires CTLA-4 (Sojka, 2009; Winstead 2008). Therefore, we showed the effect of Rap1-deficiency on the expression and function of CTLA-4. Rap1-deficiency also caused profound reduction in IL-10 production by Treg cells in the LILP (Fig. 4c). Our data suggest that Rap1-deficiency did not specifically impair CTLA-4 function, but TCR-signal-dependent functions of Treg cells. As CTLA-4 is induced in TCR-signal-dependent manner, effector Treg cells express CTLA-4 more than naïve Treg cells. ROR γ ⁺ Treg cells are effector Treg cells, and all express CTLA-4. CTLA-4⁻ Treg cells might be naïve ROR γ ⁻ Treg cells.

The expression level of CTLA-4 in Rap1-deficient Treg cells and the number are low compared with those of WT Treg cells, it is difficult to isolate WT and Rap1-deficient Treg cells expressing similar level of CTLA-4. In addition, Rap1-deficiency markedly affects integrin-dependent adhesion/migration and homing capacity, which would complicate adoptive transfer experiments.

Reviewer #2:

1. Fig 1 Authors show increased IL-17 and IL-17/IFN γ producing T cells in the colon of Rap1 ko mice, which is completely prevented by antibiotics treatment. Ko mice develop colitis which is ameliorated by antibiotics treatment. Ko of IL-17A also ameliorates colitis but Th17 development (ROR γ T⁺ T cells) was maintained.

Please provide example dot plots of ROR γ T staining within Teffector and Tregs as well as quantification of cell numbers in LP T cells. Spontaneous colitis development and Th17 accumulation has been described by the group previously. The figure is not really essential and can be combined with results from figure 3 addressing the role of Treg differentiation.

(R) As indicated, we demonstrated representative profiles of ROR γ T staining within effector T cells and Treg (Foxp3⁺) cells, and cell numbers (Fig.1a, Fig.2a and Supplementary Fig.2a).

Previous paper described the development of colitis and increased proportion of Th17 in Rap1KO mice, but did not show their causal relationship. Fig.1 is a fundamental finding of this paper demonstrating that overproduction of IL-17A is a cause of colitis in Rap1KO mice.

As indicated, we changed the order of Figures, and continued previous Fig.3 (revised Fig.2) to Fig.1.

2. Fig.2 In vitro Th17 differentiation is not altered regarding ROR γ T expression but Rap1 ko cells have less IL-17.

Under Treg inducing conditions the authors claim reduced Foxp3 induction with low CD3 stimulation but increased Foxp3 with strong CD3 signal due to a lack of apoptosis in ko versus WT cells. These Treg data are not convincing and largely unclear: the gates for Foxp3 expression cut right through the population, the effect of the culture conditions on expansion/apoptosis are unclear since no data are provided, e.g. absolute cell numbers, staining for apoptosis. It remains unclear to me what the in vitro conditions should indicate and what exactly happens to the cells. Also data in Th1 and Th2 cells are incomplete (cell numbers, expansion, cytokine production, ...). It remains unclear how the ko affects activation, proliferation, differentiation. Especially in light of the results from figures 5&6 addressing TCR signalling.

(R) As indicated, we provided the effects (proliferation, apoptosis, cytokine production) of Rap1-deficiency on the TCR-dependent Treg differentiation, as well as Th1 and Th2 differentiation *in vitro* in Figure 5 and Supplementary Fig.4.

3. Figure 3. Rap1 ko show decreased frequencies of LP ROR γ T Foxp3 cells, whereas ROR γ T Foxp3- are increased. This is not affected by IL-17A ko, i.e. not induced by inflammation. Transferred WT T cells preferentially develop into ROR γ T Treg and transfer prevents colitis. These data are potentially interesting pointing to an Treg intrinsic role of Rap1 in the intestine. All data should be condensed into 1 main figure rather than into suppl material.

(R) As indicated, we combined previous Fig.4c and Supplementary Fig.3 as a new Fig.3.

The data suggest that Rap1 ko Treg cannot prevent colitis. However, the transfer system CD25⁻ T cells does not allow to conclude that Tregs were induced from naïve non-Treg precursors. It would be important to show that by generating a Foxp3cre specific Rap1 ko to demonstrate an Treg intrinsic effect. Alternatively a colitis transfer model using mixed transfers of ko and WT cells can be used.

(R) In this study, we did not intend to prove that ROR γ T⁺ Treg cells were definitely generated from non-Treg precursors. We would like to suggest that defective generation of ROR γ T⁺ Treg cells is one of causes to develop the colitis in Rap1KO mice. For this end, we performed adoptive transfer of wild-type CD25⁻ CD4⁺ naïve T cells into Rap1KO mice, because naïve CD4⁺ cells generated ROR γ T⁺ Treg cells in the LILP (Pratama, JEM). As reported, wild-type ROR γ T⁺ Treg cells were effectively generated from wild-type CD25⁻ CD4⁺ naïve T cells in the LILP, and prevented the development of colitis in Rap1KO mice (Fig. 3).

Using Foxp3cre-specific Rap1KO mice, it is impossible to examine the effects of Rap1-deficiency on the generation of Treg cells from non-Treg precursors (Foxp3⁻). As Rap1 is indispensable for integrin-dependent homing of naïve T cells into peripheral LNs, Rap1 deficiency causes lymphopenia in lymphoid tissues (Ishihara, 2015). Foxp3cre-specific Rap1-deficiency specifically prevents the homing of Foxp3⁺ cells, and Foxp3⁺ cells will be few in peripheral lymph nodes, which must lead to the autoimmune diseases. In addition, we do not have Fox3-cre mice, and it takes long times to purchase that mice and mate them with Rap1a/b flox mice and CD4-cre mice.

Rap1-deficiency markedly affects integrin-dependent adhesion/migration and homing capacity of T cells, which would complicate adoptive transfer experiments.

I also find the gating in figure 4c not ideal: gate on all transferred or resident CD4 T cells – display Foxp3 versus ROR γ T (to visualize the Treg/Tcon ratio) second gate on Foxp3 pos or neg: display GATA3 (or Helios) versus ROR γ T and another dot plot for cytokines by both subsets.

(R) As suggested, we gated on CD4⁺-displayed Foxp3 versus ROR γ T, and second gate on Foxp3⁺ or Foxp3⁻-displayed ROR γ T⁺ or ⁻ (Fig.2 and 3) (6 colors were the limit in our flow cytometer. We confirmed that ROR γ T⁻ Treg were almost Helios⁺).

4. In figure 4 the authors show that Treg from Rap1 ko have reduced capacity to suppress to reduced CTLA4 expression and transendocytosis in response to TCR but not PMA/iono stimulation. It is not clear to me how this has been done, gating and calculations.

4 b) & d) suppression assay and transendocytosis assay: how have the Tregs been sorted, from which source and is the purity comparable between ko and WT? 4c) staining and gating of DC subsets?

(R) We presented the source and purity of Treg cells in Supplementary Fig.3b. We also showed the gating of DC in Supplementary Fig.3a.

5. In figure 5 a TCR signaling defect is demonstrated which affects T cell activation, proliferation and apoptosis in response to titrated amounts TCR stimulus. These data should be combined with data from figure 2. The overall message of these results for the interpretation of the ko phenotype are not clear to me.

(R) As indicated, we continued previous Fig.2 (revised Fig.5) and Fig.6.

6. In figure 6 the authors suggest that Rap1 ko T cells have a defect in TCR actin formation. This is not my expertise and I cannot judge the quality of the data.

However, I have the impression that for data in figure 5&6 it is not clear how this affects Tregs and how these data are connected to the *in vitro* and *in vivo* Treg phenotype and the development of colitis.

(R) Rap1-deficiency impaired not only Treg differentiation, but also Th1 and Th2 differentiation *in vitro* (Fig.5 and Supplementary Fig.4). Using PMA and ionomycin, instead of anti-CD3, there was no difference in Treg differentiation between WT and Rap1-deficient naïve CD4⁺ T cells (Fig.5). These data suggest that Rap1 is not specifically involved in Treg differentiation, but TCR-signal is impaired in Rap1-deficient cells. TCR signals are critical for Treg differentiation, Foxp3-mediated gene regulation and suppressor functions (Vahl, 2014; Levine, 2014; Ono, 2020; Li, 2016). Previous papers suggest that strong TCR signaling, especially PLC γ and Ca²⁺ signaling, is necessary for Treg cell development, maintenance or function (Moran, 2011; Koonpaew, 2006; Wu, 2006; Vaeth, 2019 #292). WASP-dependent actin foci

formation was reported to be essential for TCR-distal signals such as PLC γ and NFAT activation (Kumari, 2015). WASP was reported to be critical for peripheral Treg differentiation (Humblet-Baron, 2007).

ROR γ ⁺Treg are predominantly generated from naïve CD4⁺ cells in LILP by 4 weeks of age in a TCR-dependent manner in response to intestinal microbiota, and suppress microbe-specific Th17 differentiation (Lathrop, 2011; Russler-Germain, 2017; Sefik, 2015; Yang 2016; Xu, 2018; Neumann, 2019). Reduction in TCR-signal might impair preceding differentiation of ROR γ ⁺ Treg in LILP, which might lead to the expansion of Th17 cells and intestinal inflammation (the model is shown in Supplementary Fig.6). Intestinal inflammation impaired epithelial barrier functions and dysbiosis (Supplementary Fig.1c). Dysbiosis might accelerate IL-17A production (Xu, M., 2018; Miyauchi, E., 2020). In addition, lymphopenia was reported to induce colitogenic Th17 cells in the absence of Treg in microbiota-dependent manner (Leppkes, 2009; Feng, 2009; Kawabe, 2013) (Supplementary Fig.6)(p12, lines 1-32; p13, lines 6-18).

Reviewer #3:

One is regarding the role of the integrin adhesion receptor LFA-1. Rap-1 and its substrates are important mediators of LFA-1 conformational activation and the authors are eminent experts of deciphering molecular regulation of adhesion receptors in T cells. However, they merely cite older work of others to claim that the new Rap-1 functions described here are LFA-1-independent. I think this needs further clarification and should at least be discussed more precisely and to the point. LFA-1 is such an important player in immune synapse formation – this needs more illumination, particularly by these authors.

(R) Our previous paper (Ishihara, 2015) reported that LFA-1 regulation by Rap1 causes the colitis. In this paper, we would like to show LFA-1-independent important function of Rap1 in immune regulation. As pointed, integrins play central roles in many aspects of the development of colitis in Rap1KO mice. We added the statement of the roles of Rap1 as the integrin regulator in the development of colitis in the discussion section (p12, lines 1-12; p13, lines 6-18).

The other concern is regarding the mechanistic analyses part of the paper. They are novel and interesting, but remain somewhat descriptive and do not really explain a lot.

1) Where does the specificity of Rap-1 function in Treg regulation and more selectively in ROR γ ⁺Treg differentiation come from?

(R) Rap1-deficiency impaired not only Treg differentiation, but also Th1 and Th2 differentiation *in vitro* (Fig.5 and Supplementary Fig.4). Using PMA and ionophore, instead of anti-CD3, there was no difference in Treg differentiation between WT and Rap1-deficient naïve CD4⁺ T cells (Fig.5). These data suggest that Rap1 is not specifically involved in Treg differentiation, but TCR-signal is impaired in Rap1-deficient cells. TCR signals are critical for Treg differentiation, Foxp3-mediated gene regulation and suppressor functions (Vahl, 2014; Levine, 2014; Ono, 2020; Li, 2016). Previous papers suggest that strong TCR signaling, especially PLC γ and Ca²⁺ signaling, is necessary for Treg cell development, maintenance or function (Moran, 2011; Koonpaew, 2006; Wu, 2006; Vaeth, 2019 #292). ROR γ ⁺Treg are predominantly generated from naïve CD4⁺ cells in LILP (large intestine lamina propria) by 4 weeks of age in a TCR-dependent manner in response to intestinal microbiota, and suppress the microbe-specific Th17 differentiation (Lathrop, 2011; Russler-Germain, 2017; Sefik, 2015; Yang 2016; Xu, 2018; Neumann, 2019). Reduction in TCR-signal impairs the preceding differentiation of ROR γ ⁺Treg in the LILP of Rap1KO mice, therefore

lymphopenia might accelerate the generation and expansion of Th17 cells in absence of ROR γ ⁺ Treg. We described those explanations in the discussion section (p12, lines 13-27) (the model is shown in Supplementary Fig.6).

According to the authors, this is due to downregulation of CTLA-4 expression and the resultant lack of CD80/86 transendocytosis? How could this work? Is CTLA-4 selectively associated or co-localized with Rap-1 or its substrates?

Also, and importantly: if CTLA-4 really is the direct and specific target of Rap-1 function in this context, then the regulatory impact of Rap-1 should not be restricted to Tregs, because CTLA-4 is not a Treg-specific molecule. How about CTLA-4 expression and CD80/86 transendocytosis on Rap-1^{-/-} effector cells, including IL17 expressing cells?

And with respect to Tregs: does Rap1 deficiency affect Foxp3 expression (via NFAT inactivation)? What is the relation of actin foci regulation with respect to CTLA-4 regulation and CD80/86 transendocytosis?

(R) We did not insist that Rap1-deficiency specifically impaired CTLA-4 function. The CTLA-4 pathways are reported to represent the core mechanism of Treg cell suppression that is indispensable for immune homeostasis (Wing, 2008). In addition, the inhibition of CD4⁺ T cell expansion by Treg in a lymphopenic environment requires CTLA-4 (Sojka, 2009; Winstead 2008). Therefore, we showed the effects of Rap1-deficiency on the expression and function of CTLA-4. However, Rap1-deficiency reduced TCR-dependent, but not PMA-ionomycin-dependent CTLA-4 induction and trans-endocytosis of CD80/86 in naïve CD4⁺ T cells (Fig. 5c). Rap1-deficiency also caused profound reduction in IL-10 production by Treg cells in the LILP (Fig. 4c). These data suggest that Rap1-deficiency did not directly impair CTLA-4 function, but defective TCR-signals reduced the expression of CTLA-4.

Rap1-deficient CD4⁺ cells demonstrated reduced TCR-signals and TCR-dependent expression of Foxp3 and CTLA-4 *in vitro* (Fig.5 and 6). We found that Rap1-deficiency caused a defect in the formation of actin foci (Fig.7). WASP-dependent actin foci formation was reported to be important for TCR signals such as PLC γ and NFAT activation (Kumari, 2015). WASP was reported to be critical for peripheral Treg differentiation (Foxp3 induction) (Humblet-Baron, 2007). As pointed, although we could not prove the relationship between CTLA-4 expression and actin foci formation, Rap1-dependent TCR-signals might be critical for the Foxp3 expression and Foxp3-dependent functions (such as CTLA-4 and IL-10) through the regulation of both LFA-1 and TCR-dependent adhesion with APC *in vivo* (p13, lines 6-18).

How does the WASP association of Rap-1 aid in this?

I can certainly see that the whole workup of this issue would result in substantial more experimentation, but the presented explanations and associations are nonetheless rather loose and vague.

(R) We have explored the downstream effector molecules of Rap1 to regulate WASP-dependent actin foci. RIAM is a possible candidate of actin regulator to associate directly with Rap1-GTP and be involved in TCR-dependent PLC γ localization (Lafuente, 2006; Wynne, 2012). However, RIAM and Lpd were not observed in actin foci, different from Rap1-GTP (Supplementary Fig.5b). We also generated RIAM-knockout 3A9T cells by crisper/cas9 system, and examined the effect of RIAM-deficiency on actin foci. RIAMKO cells showed a different morphology, but formed actin foci by anti-CD3 stimulation, although Rap1a/b-knockdown cells demonstrated the reduced formation of actin foci (Supplementary Fig.5c). The binding of Rap1-GDP/GTP to WASP suggest that Rap1 might directly or indirectly be necessary for the recruitment of WASP into MC. We added those data in revised Supplementary Fig.5 and improved the discussion about the mechanism of actin foci regulation by Rap1 (p14, lines 3-11).

Minor point:

The classification “Rap1 is involved in distal TCR signaling” seems rather arbitrary. If this relates to a temporal axis, then Rap1-deficiency does not seem to alter phosphorylation responses (Fig 5d) over extended time courses (hours), but it does affect calcium signaling on a rather short time scale (peaks roughly at 10 min after stimulation). The term should thus be defined more precisely or changed.

(R) We deleted [distal]TCR signaling.

Reviewer #4:

why loss of Rap1 specifically affects Treg cells. If this molecule was downstream of TCR signalling in all T cells (CD4/CD8), these mice would also have defect in Th17 cells, which seems not to be the case *in vivo*.

1. Quantification (graph) for Fig 1b.

(R) As indicated, we quantified the data in Fig.1b.

2. Fig1c. Spellcheck “antibiotics”

(R) We corrected the wrong spell.

3. In the results section, the authors should mention that Rap1KO is a T cell specific deletion of Rap1a/b using CD4-cre. Without reading the methods section, this aspect is not obvious.

(R) As indicated, we described Rap1KO mice have a T cell specific deletion of Rap1a/b using CD4-cre mice (p5, lines 4-5).

4. In Fig 2b, gating for Foxp3+ cells should be relaxed. The horizontal line in the quadrant could be lowered to include all Foxp3 expressing cells.

(R) As suggested, we have carefully determined the horizontal line for gating of Foxp3⁺ (Fig.5b).

5. In low anti-CD3 Treg polarizing condition, if the gating is adjusted, %Foxp3+ cells might become similar in WT and Rap1Kos. However, there appears to be a clear defect in Foxp3 MFI in the KOs.

(R) As pointed, Foxp3 MFI was significantly low in Rap1KO mice when stimulated in Treg polarizing condition in Fig.5b.

6. While CD4+ T cells in the intestine, blood and mesLNs show more IL-17A expression in Rap1KOs, this phenotype seems to be reversed *in vitro*. The authors should discuss this. Is it simply the lack of Tregs?

(R) As Rap1 is indispensable for integrin-dependent homing of naïve T cells into peripheral LNs, Rap1 deficiency causes lymphopenia in mesenteric lymph nodes and

LILP (Ishihara, 2015). Lymphopenia-induced spontaneous proliferation of naïve T cells generates colitogenic Th17 cells in the absence of Treg cells in a microbiota-dependent manner (Leppkes, 2009; Feng, 2010; Kawabe, 2013).

TCR signals are critical for Treg differentiation, Foxp3-mediated gene regulation and suppressor functions (Vahl, 2014; Levine, 2014; Ono, 2020; Li, 2016). Previous papers suggest that strong TCR signaling, especially PLC γ and Ca²⁺ signaling, is necessary for Treg cell development, maintenance or function (Moran, 2011; Koonpaew, 2006; Wu, 2006; Vaeth, 2019 #292). ROR γ ⁺ Treg are predominantly generated from naïve CD4⁺ cells in LILP (large intestine lamina propria) by 4 weeks of age in a TCR-dependent manner in response to intestinal microbiota, and suppress microbe-specific Th17 differentiation (Lathrop, 2011; Russler-Germain, 2017; Sefik, 2015; Yang 2016; Xu, 2018; Neumann, 2019). ROR γ ⁺ Treg cell deficiency is reported to exacerbate colitogenic Th17 cell differentiation and promotes colonic inflammation. Reduction in TCR-signal might impair preceding differentiation of ROR γ ⁺ Treg in LILP.

Thus, lymphopenia and defective Treg functions might lead to the expansion of Th17 cells and intestinal inflammation. Intestinal inflammation impaired epithelial barrier functions and dysbiosis (Supplementary Fig. 1c), which might accelerate IL-17A production (Xu, M., 2018; Miyauchi, E., 2020). We showed the model in Supplementary Fig. 6.

7. Is Rap1 a Treg specific molecule? Why does its loss affect Rorgt induction in Tregs but not in CD4 T cells? Or is it a defect in Foxp3 induction in vivo?

(R) ROR γ ⁺ Treg cells are generated from naïve CD4⁺ T cells, but not from ROR γ ⁻ Treg cells in LILP (large intestine lamina propria) in a TCR-dependent manner in response to intestinal microbiota (Lathrop, 2011; Russler-Germain, 2017; Sefik, 2015; Xu, 2018; Neumann, 2019). ROR γ ⁻ Treg cells are differentiated in the thymus, and do not express ROR γ in LILP.

Rap1-deficient naïve CD4⁺ cells demonstrated a reduction in TCR-signals (NFAT activation etc) (Fig. 6), and the impaired expression of Foxp3 *in vitro* (Fig. 5). As described above, TCR signals are critical for Foxp3 expression, Foxp3-mediated gene regulation and suppressor functions.

As suggested, a defect in TCR-dependent Foxp3 induction might reduce the generation of ROR γ ⁺ Foxp3⁺ cells in the LILP of Rap1KO mice (p12, lines 12-24).

8. To understand the differential role of Rap1 in Treg cells and Foxp3-CD4 T cells, the authors should compare the proliferation of Rap1 deficient Tregs and CD4 T cells to WT counterparts.

(R) As described above, our data suggest that Rap1-deficiency impairs TCR-dependent functions in both Treg and Tconv cells. We showed TCR-induced proliferation of WT and Rap1-deficient Treg cells at 0.5 μ g/ml of anti-CD3 (Supplementary Fig.3b), although naïve Rap1-deficient Treg cells are few and it is difficult to isolate them.

9. Fig 5d. Western blot needs loading control.

(R) As pointed, we added the loading control (actin).

REVIEWERS' COMMENTS:

Reviewer #1 (Remarks to the Author):

The authors addressed my questions/ concerns to the most part. Although I would have liked to see post-antibiotic treatment microbial diversity by 16s, the study in my opinion could now be published,

Reviewer #2 (Remarks to the Author):

The authors have significantly improved their manuscript and addressed my main technical concerns.

However, I still have difficulties to grasp the main message of the paper. To me it is confusing the way it is structured and I think this could now be better worked out.

In my view the central aspect is the effect on TCR signalling (now at the end), which mainly affects Treg development and functionality in vivo, despite the fact that it is not a Treg specific defect.

This should be at the beginning allowing the reader to clearly understand that the remainder of the paper is to find out how the defective TCR signalling translates in such specific in vivo findings.

All other in vivo findings are downstream of the TCR signaling defect, e.g. block of Treg-mediated Th17 control. This Th17 selectivity is most likely due to the fact that no Th1 or Th2 biased disease models were used, since it appears that in vitro all Th subsets are impaired.

Also the title could be more specific highlighting the main effect on TCR signalling. e.g. (just a quick suggestion)

"Role of Rap1 in TCR signalling affecting Treg development and Treg-mediated control of intestinal colitogenic Th17 responses

Reviewer #3 (Remarks to the Author):

To me, the clarifications given by the authors are sufficient. In fact, they are essential to not misinterpret the data in the sense that Rap1 does not selectively impair TCR signaling in Tregs, but that the impairment of Treg function/differentiation in Rap1 ko mice is the limiting issue for the colitis context which is analyzed here. The new clarifications in the text address this thoroughly, as said before. However, I would suggest that they state this very clearly in the abstract as well. This is not an essential point, but it would help to put the key message of the paper in context for the reader from the beginning. Otherwise, I see all of my points addressed and the current version of the manuscript.

Reviewer #4 (Remarks to the Author):

The authors have addressed all the concerns of this reviewer. Congratulations on this beautiful work!!